# Concurrent decoding of distinct neurophysiological fingerprints of tremor and bradykinesia in Parkinson's disease

**Peter M Lauro**[1,2,3]*, **Shane Lee**[1,2,4,5], **Daniel E Amaya**[1,2], **David D Liu**[6], **Umer Akbar**[2,3,4,7], **Wael F Asaad**[1,2,3,4,5]*

[1]Department of Neuroscience, Brown University, Providence, United States; [2]Robert J. and Nancy D. Carney Institute for Brain Science, Brown University, Providence, United States; [3]The Warren Alpert Medical School, Brown University, Providence, United States; [4]Norman Prince Neurosciences Institute, Rhode Island Hospital, Providence, United States; [5]Department of Neurosurgery, Rhode Island Hospital, Providence, United States; [6]Department of Neurosurgery, Brigham and Women's Hospital, Boston, United States; [7]Department of Neurology, Rhode Island Hospital, Providence, United States

**\*For correspondence:**
me@peterlauro.me (PML);
Wael_Asaad@brown.edu (WFA)

**Abstract** Parkinson's disease (PD) is characterized by distinct motor phenomena that are expressed asynchronously. Understanding the neurophysiological correlates of these motor states could facilitate monitoring of disease progression and allow improved assessments of therapeutic efficacy, as well as enable optimal closed-loop neuromodulation. We examined neural activity in the basal ganglia and cortex of 31 subjects with PD during a quantitative motor task to decode tremor and bradykinesia – two cardinal motor signs of PD – and relatively asymptomatic periods of behavior. Support vector regression analysis of microelectrode and electrocorticography recordings revealed that tremor and bradykinesia had nearly opposite neural signatures, while effective motor control displayed unique, differentiating features. The neurophysiological signatures of these motor states depended on the signal type and location. Cortical decoding generally outperformed subcortical decoding. Within the subthalamic nucleus (STN), tremor and bradykinesia were better decoded from distinct subregions. These results demonstrate how to leverage neurophysiology to more precisely treat PD.

## Editor's evaluation

This important study advances our understanding of Parkinson's by identifying micro and macro scale signatures linked to critical symptoms (e.g., tremor and slowness of movement), and effective motor control. The evidence supporting the conclusions is solid, and leverages a rich dataset obtained during naturalistic movement. The work will be of interest to neuroscientists, neurologists, and biomedical engineers.

## Introduction

Parkinson's disease (PD) is a common and complex neurodegenerative disorder characterized by the dynamic expression of particular motor features such as tremor and bradykinesia (*Armstrong and Okun, 2020*; *Parkinson, 2002*). These distinct motor signs are expressed variably across patients and may respond differently to dopamine replacement therapy; their differential expression is often used to classify patients into phenotypic subtypes (*Koller, 1986*; *Sethi, 2008*). Despite this heterogeneity,

both of these motor features (and both tremor-dominant and non-tremor-dominant patient subtypes) respond to high-frequency deep brain stimulation (DBS) applied to the subthalamic nucleus (STN) (*Katz et al., 2015*; *Limousin et al., 1998*).

DBS delivered in a closed-loop fashion (i.e., in response to neurophysiological biomarkers) has shown promising therapeutic potential primarily toward alleviating bradykinesia (*Little et al., 2016a*; *Little et al., 2016b*), but current efforts focusing on $\beta$ frequency oscillations (15–30 Hz) have been shown to inadequately treat or worsen tremor in some cases (*Piña-Fuentes et al., 2020*; *Velisar et al., 2019*). Thus, tremor may be better signaled by different components within the local field potential (LFP) spectrum, and closed-loop DBS could benefit from a clearer understanding of the neurophysiological biomarkers that differentiate these motor signs from each other, and from more optimal motor performance in the absence of these impairments.

To this point, STN LFP recordings from patients with different PD subtypes have revealed distinct patterns of oscillatory activity (*Telkes et al., 2018*). In addition to spectral variability, specific stimulation sites within the STN have been associated with the preferential reduction of individual motor signs (*Akram et al., 2017*). Moreover, these STN sites were associated with specific patterns of anatomical connectivity with cortical structures (*Haynes and Haber, 2013*). Much like how overlapping subdivisions of basal-ganglia-cortical circuits have been found to encode separate aspects of movement (*Mosher et al., 2021*; *Neumann et al., 2018*), separate motor features may be mediated by different sub-circuits involving the STN and sensorimotor cortex (*Gibson et al., 2021*).

In order to better reveal the functional and anatomical substrates of distinct PD motor states, we enlisted patients with PD undergoing awake DBS electrode implantation to perform a continuous visual-motor task that allowed rigorous, concurrent measurement of different motor metrics while we acquired STN (micro- and macroelectrode) and cortical (electrocorticography [ECoG]) recordings. Prior studies have not attempted to simultaneously decode different aspects of disease expression, contrast these measures with symptom-free performance, and examine disease expression on the short timescales relevant to that varying expression. While our group has previously demonstrated the ability to decode global PD motor dysfunction from STN recordings on short timescales (*Ahn et al., 2020*; *Sanderson et al., 2020*), we focus here on individual motor features and their specific neurophysiological manifestations. Specifically, we trained machine learning models to directly decode tremor or slowness from neural recordings to reveal the spectral and anatomical fingerprints of these cardinal motor features of PD.

## Results

### Motor behavior during the target tracking task

Twenty-seven patients with PD undergoing STN DBS implantation and 17 age-matched controls performed a visual-motor task in which they followed an on-screen target with a cursor controlled by either a joystick or a stylus and tablet (*Figure 1A*). Twenty-three patients (and 12 control subjects) performed a version of the task with fixed patterns of target movement, while four patients (and five control subjects) performed a version with randomly generated target paths. Each patient performed 1–4 sessions of the task during the procedure for a total of 69 sessions, while control subjects each performed 1 session extra-operatively for a total of 17 sessions. Tremor amplitude and cursor speed – task metrics calculated to reflect the expression of tremor and bradykinesia – were quantified from the cursor traces. These behavioral metric data were then averaged into 7 s non-overlapping epochs. To compare metrics across subject populations while considering epochs, trials, and sessions as repeated measurements within individuals, linear mixed models (LMMs) were used (see Materials and methods). The resulting metric distributions for PD vs. control subjects demonstrated increased tremor for PD patients (*n*=6498 epochs across 44 subjects, PD vs. control, LMM coefficient/LMM $\beta$=0.337, *Z*=2.169, *p*=0.030), but only a trend for decreased speed (PD vs. control, LMM $\beta$=−0.592, *Z*=−1.194, *p*=0.232) (*Figure 1B and C*).

Tremor distributions compiled across task versions revealed that while subjects with PD spent a substantial fraction of time without tremor, they also exhibited a large range of tremor expression not present in control subjects (*Figure 1B and C*, top). On the other hand, the two task versions generated different movement speed distributions (*Figure 1B and C*, bottom). While the fixed-pattern version of the task elicited a bimodal distribution (reflecting slower turns and faster straight path segments) in

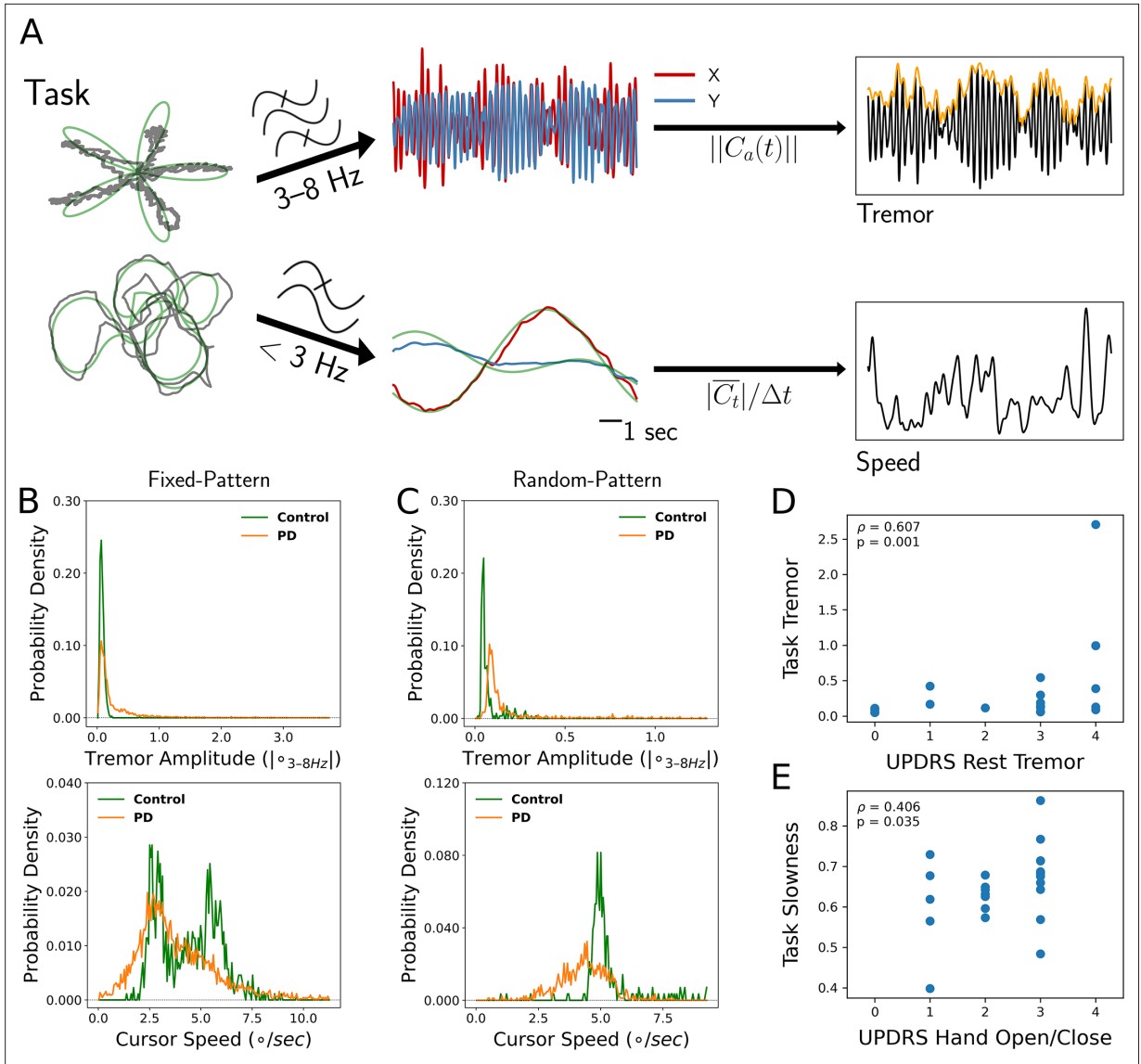

**Figure 1.** Tremor and movement speed calculated from fixed- and random-pattern intraoperative visual-motor tasks. (**A**) Left: Schematic of task target (green) and cursor (gray) traces from a single trial of the fixed- (top) or random- (bottom) pattern task. Center-top: Bandpass filtered cursor traces from a task trial. $||C_a||$ refers to the amplitude of the analytic signal (**a**) of the cursor trace (**C**). Center-bottom: Lowpass filtered cursor traces from a task trial. Right-top: One-dimensional projection of bandpass filtered traces (black), with tremor amplitude measured from the envelope (orange). Right-bottom: Cursor speed measured from lowpass filtered traces (black). Figure adapted from Figure 1 of **Ahn et al., 2020**. (**B, C**) Distributions of 7 s tremor amplitude (top) and cursor speed (bottom) epochs for control subject and Parkinson's disease (PD) patient populations in the fixed-pattern (**B**) (*n*=5375 epochs across 35 subjects) and random-pattern (**C**) (*n*=1123 epochs across 9 subjects) task. ° – degrees of visual angle. (**D, E**) Task-based tremor amplitude (**D**) and slowness (**E**) corresponded to UPDRS measures of tremor (**D**) or bradykinesia (**E**) (*n*=24 subjects). *ρ*=Spearman correlation statistic.

control subjects, the random-pattern version elicited a single peak corresponding to the fixed target speed used in that task. Nonetheless, in the random-pattern version, the PD cursor speed distribution was shifted to the left (i.e., slower) relative to control subjects (*n*=1123 epochs across 9 subjects, PD vs. control, LMM $\beta$=−1.004, *Z*=−2.210, *p*=0.027). Cursor speed was converted to 'slowness' in order to control for target trajectory/speed variability by normalizing each session's distribution of cursor speed to its minimum and maximum values (0: highest speed, 1: lowest speed). To determine whether these two motor metrics reflected distinct components of PD motor dysfunction, each patient's metric distribution medians were correlated against their UPDRS III motor subscores. Indeed, tremor amplitude positively correlated with the resting tremor subscore ( $\rho$ =0.607, *p*=0.001, *n*=24 PD subjects, Spearman correlation) (**Figure 1D**) and slowness positively correlated with the hand open/

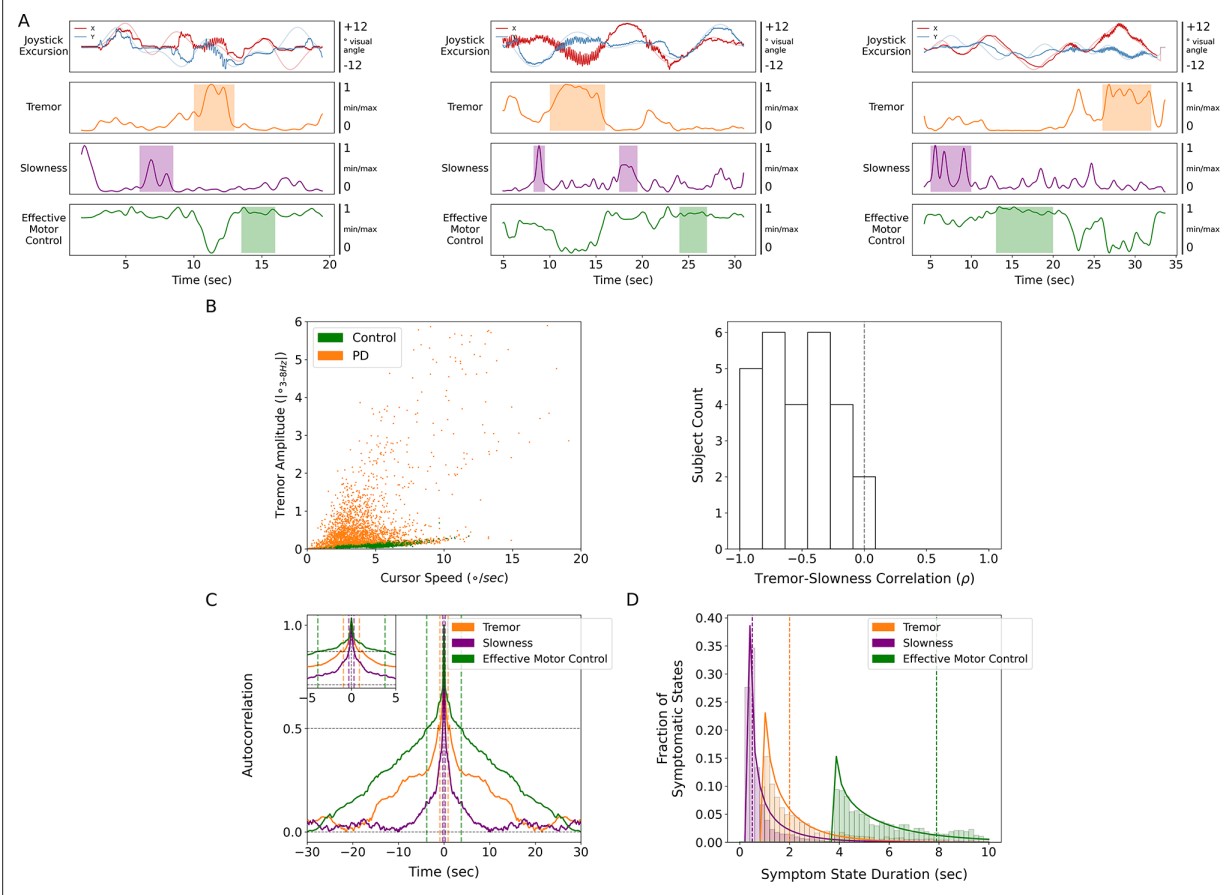

**Figure 2.** Tremor and slowness represented two non-overlapping motor states with differing timescales. (**A**) Examples from three individual subjects of cursor (solid lines) and target (translucent lines) traces (top row) and calculated motor metrics (bottom three rows) within single trials. Periods of increased expression of individual motor metrics are highlighted by their respective color. (**B**) (Left) Scatter plot of all cursor speed and tremor measurements in 7 s epochs across subjects. (Right) Histogram of subject-wide behavioral Spearman correlation with tremor and slowness metrics (*n*=27 subjects). (**C**) Autocorrelograms of symptomatic (tremor, slowness) and non-symptomatic (effective motor control) metrics. Colored vertical dashed lines indicate full-width half-maximum (FWHM) for each metric. Top-left inset depicts a zoomed-in window of the autocorrelogram. (**D**) Histogram of sustained motor metric period duration (i.e., symptomatic state duration) across subjects. Solid lines indicate gamma distribution fit to each motor metric state histogram, while dashed vertical lines indicate the median symptomatic state length for each metric.

close subscore (a subscore used in part to assess bradykinesia) ( $\rho$ =0.406, *p*=0.035, Spearman correlation) (*Figure 1E*). However, the opposite correlations were not significant (tremor amplitude – UPDRS hand open/close: $\rho$ =0.022, *p*=0.446; slowness – UPDRS resting tremor: $\rho$ =0.120, *p*=0.219). In addition, action/postural tremor subscores trended toward a positive correlation with tremor amplitude ( $\rho$ =0.354, *p*=0.057), while resting and action/postural tremor subscores were positively correlated within subjects ( $\rho$ =0.602, *p*=0.002). Although postural tremor can correlate with resting tremor when patients with PD are measured by the UPDRS for the former and the Washington Heights-Inwood Genetic Study of Essential Tremor (WHIGET) Rating Scale for the latter, resting tremor is thought to be more specific to the PD pathophysiology (*Louis et al., 2001*). Therefore, tremor and movement speed were considered to reflect two key aspects of PD motor dysfunction.

## Tremor and slowness were distinct and opposing symptomatic states

Relative to each other, tremor and slowness typically did not co-occur but rather were inversely expressed in time (*n*=27 subjects, tremor × slowness, LMM $\beta$=−0.584, *Z*=−19.351, *p*=2.00*10⁻⁸³) (*Figure 2A and B*). To understand whether this anti-correlation may have been due in part to motor features manifesting on different timescales, autocorrelograms were computed for each metric with 100 ms epochs. Here, tremor was typically expressed continuously for longer periods (autocorrelogram full-width half-maximum [FWHM], 0.898 s) as opposed to slowness (0.297 s) (*Figure 2C*). Using

this FWHM as the minimum, we calculated periods of time where metrics were sustained above control levels (i.e., symptomatic periods) across subjects. The median duration of symptomatic tremor episodes was 2.000 s, and slowness episodes lasted for 0.500 s (*Figure 2D*). With these differing timescales and anti-correlated presence, tremor and slowness appeared to represent distinct symptomatic states.

Because PD produces a fluctuating motor deficit such that there can be moments of normal-appearing motor behavior (*Mazzoni et al., 2007*), we labeled epochs without motor dysfunction as 'effective' motor states. Specifically, epochs with lower tremor and/or higher movement speeds were assigned values closer to 1 while more symptomatic epochs (high tremor and/or slower movement speeds) were assigned values closer to 0. Compared to other metrics, effective motor control was expressed on longer timescales (FWHM = 3.784 s, median state length = 7.900 s) (*Figure 2A, C and D*).

## Tremor and slowness had distinct representations within the STN

A total of 203 microelectrode and 176 macroelectrode recordings (microelectrode tips and macroelectrode contacts separated by 3 mm on the same electrodes) were acquired from the STN as patients performed the task. To assess whether tremor or slowness could be decoded from these recordings, spectral estimates of power from 3 to 400 Hz were obtained using a wavelet convolution. Narrowband power estimates were grouped into six broad frequency bands ($\theta/\alpha, \beta, \gamma_{low}, \gamma_{mid}, \gamma_{high}, hfo$) with 7 sub-bands each, for a total of 42 neural 'features' per 7 s epoch (*Ahn et al., 2020*). Neural decoding models (support vector regression [SVR] with a linear kernel and 100-fold cross-validation) were trained directly on the epoch's average metric (tremor or slowness values averaged within each epoch), and their performance was assessed with squared Pearson's $r$ ($r^2$) between observed and decoded metrics. Across each subject's best-performing microelectrode recordings (MER), tremor decoding performance ($r^2 = 0.232 \pm 0.200$) was superior to slowness decoding ($r^2 = 0.125 \pm 0.108$) (n=203 MER models, tremor v. slowness, LMM $\beta$=0.051, Z=5.477, p=4.33*10$^{-8}$). No such difference was observed across macroelectrode recordings (n=176 macroelectrode recordings models, tremor v. slowness, $r^2 = 0.209 \pm 0.174$ v. $r^2 = 0.198 \pm 0.147$, LMM $\beta$=−0.005, Z=−0.496, p=0.620). To determine if tremor and slowness had distinct neurophysiological signatures, SVR model feature weights were aggregated for each metric. To understand which spectral features were used consistently across models, feature weights were compared to null distributions generated from models where motor metric values were shuffled with respect to the corresponding spectral features. Microelectrode tremor decoding models positively weighted low-frequency features ($\theta, \alpha, \beta$; 4–21 Hz; p<0.001, permutation test, see Materials and methods). *Hfo* (275–375 Hz) weights were also positively associated with tremor decoding (p<0.014, permutation test). In contrast, macroelectrode tremor decoding models negatively weighed $\beta$ power (14–41 Hz; p<0.026, permutation test) while positively weighing $\gamma/hfo$ activity (60–375 Hz; p<0.011, permutation test). In other words, optimal macroelectrode tremor decoding relied on decreased $\beta$ power and increased $\gamma/hfo$ power.

For slowness, microelectrode decoding models had negative $\theta$, $\gamma_{low}$, and *hfo* weights (5–12 Hz, 33–56 Hz, 200–375 Hz) (p<0.012, permutation test). Macroelectrode decoding models positively weighted $\beta$ frequencies (12–30 Hz; p<0.006, permutation test) along with negative $\gamma/hfo$ weights (33–375 Hz; p<0.001, permutation test). Tremor and slowness model features differed when compared directly, with *hfo* frequencies (225–375 Hz) being elevated during tremor in both micro/macroelectrode recordings (*Figure 3A*). Overall, just as tremor and slowness represented two distinct, anti-correlated symptomatic states of PD, tremor and slowness decoding models from the STN revealed distinguishable patterns of underlying neural activity.

However, in order to rule out the possibility that the alternating patterns of relevant neural decoding features simply reflected the anti-correlated nature of tremor and slowness, we tested whether decoding models trained for tremor could accurately decode slowness. When directly comparing tremor and slowness decoding performance on tremor-trained models, slowness decoding was inferior for both microelectrode (tremor v. slowness decoding, $r^2$=0.232±0.197 v. 0.002±0.001; LMM $\beta$=0.101, Z=11.242, p=2.54*10$^{-29}$) and macroelectrode (tremor v. slowness decoding, $r^2$=0.205±0.172 v. 0.002±0.001; LMM $\beta$=0.086, Z=10.294, p=7.53*10$^{-25}$) recordings. If decoding features for tremor and slowness were simply inverted, applying models for decoding another metric would result in

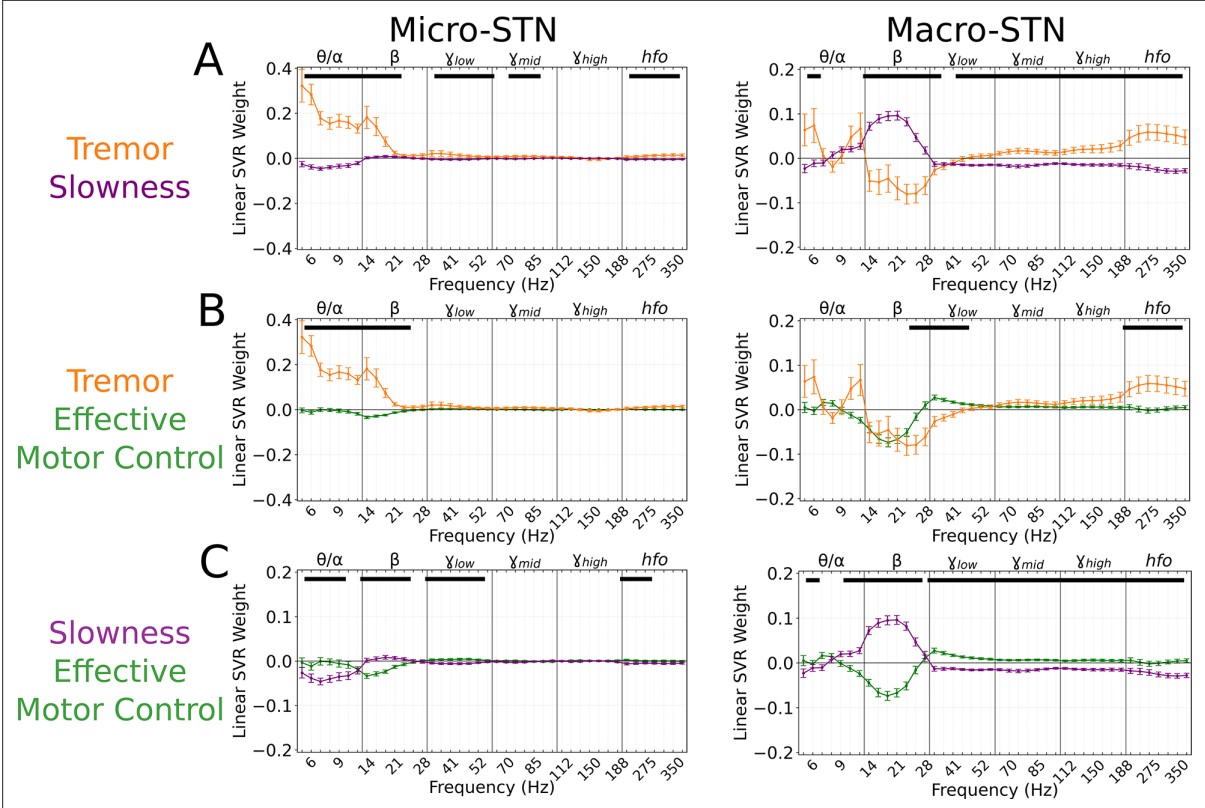

**Figure 3.** Subthalamic tremor decoding models emphasized lower frequencies whereas slowness models emphasized higher frequencies. (**A**) Average tremor decoding and slowness model coefficients for all subthalamic nucleus (STN) microelectrode (left) (*n*=203 microelectrode recordings, 27 subjects) and macroelectrode (right) recordings (*n*=176 macroelectrode recordings, 27 subjects). Solid lines indicate average weights, with positive/negative values reflecting a positive or negative relationship with the metric. Error bars indicate s.e.m. across subjects. Black lines (top) represented contiguous spectral features that significantly differed between tremor and slowness decoding models. (**B**) Average model coefficients for effective motor control and tremor for all STN microelectrode (left) and macroelectrode (right) recordings. (**C**) Average model coefficients for effective motor control and slowness for all STN microelectrode (left) and macroelectrode (right) recordings.

significantly negative $r^2$-values. Thus, the neural features used for individual metric decoding likely reflected a unique spectral state or 'fingerprint'.

To further validate that our approach was able to decode motor dysfunction in a symptom-specific fashion, we examined the relationship between individual tremor expression and tremor decoding performance (as not all patients with PD exhibit tremor). Here, we found that task-based tremor distribution medians positively correlated with individual's highest decoding performance (*n*=24 patients, $\rho$ =0.442, *p*=0.031). While UPDRS measures of tremor did not directly correlate with decoding performance (*p*<0.878) – likely due to a variety of factors distinguishing standard UPDRS tremor assessment from intraoperative task performance – the dynamic, continuous, low-velocity-biased expression of tremor on the naturalistic task (which itself did correlate with UPDRS resting tremor scores) provided an immediate, ground-truth behavior for patient-specific neurophysiological decoding.

### Effective motor control had characteristic neural signatures

Effective motor control was similarly decoded from both micro- ($r^2$=0.140±0.104) and macroelectrode ($r^2$=0.204±0.097) recordings. Effective motor control decoding was characterized by the absence of $\beta$ (10–28 Hz) power in both micro- and macroelectrode recordings (*p*<0.006, permutation test), while macroelectrodes also exhibited positive $\gamma$ power weights (30–175 Hz; *p*<0.020, permutation test). Power in $\gamma_{low}$ frequencies (30–48 Hz) in particular was significantly increased during effective motor control relative to both tremor and slowness decoding models (*p*<0.006, permutation test) (*Figure 3B–C*, right).

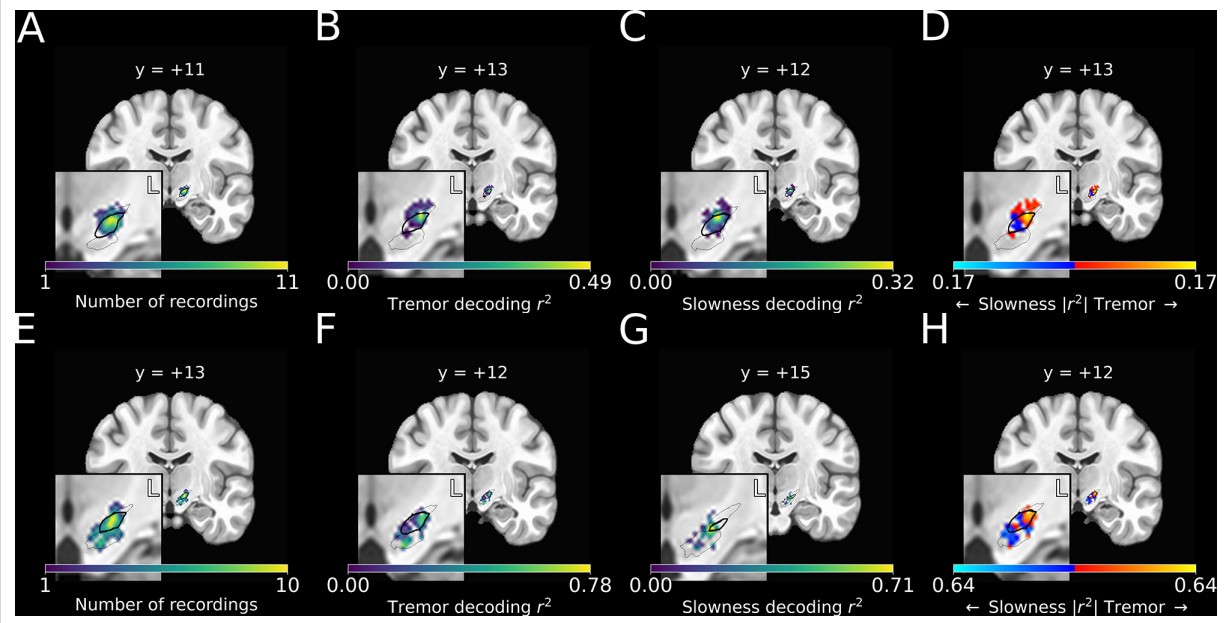

**Figure 4.** Optimal subthalamic tremor decoding sites were dorsolateral to optimal slowness decoding sites. (**A**) Recording density of stationary microelectrode recordings across patients (n=182 microelectrode recording sites, 25 subjects) and task sessions overlaid on an MNI reference volume (approximate outline of the subthalamic nucleus [STN] in bolded black, zona incerta outlined above, substantia nigra outlined below). L: left. y-value corresponds to coronal slice in MNI space. (**B**) Tremor decoding model $r^2$-values for stationary microelectrode recordings. (**C**) Slowness decoding model $r^2$-values for stationary microelectrode recordings. (**D**) Difference in tremor vs. slowness decoding $r^2$-values for stationary microelectrode recordings. Warmer colors indicate voxels where tremor decoding was superior, whereas cooler colors indicate where slowness decoding was superior. (**E**) Recording density of moving microelectrode recordings across all patients and task sessions overlaid on an MNI reference volume. (**F**) Tremor decoding model $r^2$-values for high-density STN survey recordings. (**G**) Slowness decoding model $r^2$-values for high-density STN survey recordings. (**H**) Difference in tremor vs. slowness decoding $r^2$-values for high-density STN survey recordings. $r^2$-Values depicted here are site-specific $r^2$-values generated from the whole-STN model applied to individual depth recordings. Warmer colors indicate voxels where tremor decoding was superior, whereas cooler colors indicate where slowness decoding was superior.

In total, STN activity contained specific features that distinguished symptomatic from non-symptomatic motor states. Tremor was characterized by lower frequencies ($\theta/\alpha$) in microelectrodes, slowness by $\beta$ frequencies in macroelectrodes, and effective motor performance was uniquely characterized by $\gamma_{low}$ frequencies from both recording types.

### Full-spectrum neural decoding outperformed beta-band decoding

To directly test whether each behavior model used neural features across the spectrum, we compared the relative ability of full-spectrum and canonical band ($\beta$, 12–30 Hz) models. Full-spectrum decoding had significantly greater performance for macroelectrode (full vs. beta-only decoding, LMM $\beta$=0.018–0.035, Z=2.388–3.949, p<0.017) and microelectrode (full vs. beta-only decoding, LMM $\beta$=0.014–0.017, Z=2.241–3.154, p<0.025) for all three metrics, with the exception of microelectrode-tremor decoding (full vs. beta-only decoding, LMM $\beta$=0.014, Z=1.548, p=0.122).

### Optimal subthalamic tremor decoding sites were dorsolateral to optimal slowness decoding sites across patients

To investigate whether tremor and slowness were more optimally decoded from distinct areas within the STN, recording sites for each session were reconstructed using subject-specific neuroimaging (peak MER density in MNI space: $x = -12, y = -10, z = -6.0$) (**Figure 4A**; peak macroelectrode recording density: $x = -12, y = -9, z = -3.0$). For each recording site, the corresponding decoding model performance for each metric was plotted (**Figure 4B and C**).

We then compared the voxel-wise relative performance between tremor and slowness throughout all recorded STN voxels by using a modified 3D t-test with spatially based permutation shuffling (see Materials and methods). Tremor was better decoded in recordings from dorsolateral STN (n=182

MER sites, $x = -14.0, y = -13.0, z = -5.0$; $Z = 2.116, p = 0.017$), whereas slowness was better decoded from recordings in central/ventromedial STN ($x = -12.0, y = -14.0, z = -6.0$; $Z = 1.911, p = 0.028$) (*Figure 4D*).

Optimal locations for tremor and slowness decoding were not found to differ significantly by macro-electrode location ($n$=176 macroelectrode recording sites, $p$>0.05). Moreover, the locus of optimal effective motor control decoding was not observed to differ from those of tremor or speed using either micro- and macroelectrode recordings ($p$>0.05). Nevertheless, we found that that differences in metric decoding were not only related to the frequencies present, but also to an electrode's location within the STN, as assessed over the entire study PD population.

## Optimal subthalamic tremor decoding sites were dorsolateral to optimal slowness decoding sites within individual patients

To verify the spatial relationship of optimal tremor and slowness decoding within patients, five additional right-handed patients (70.0±8.9 years of age; 2F, 3M; UPDRS III: 45.2±9.5) underwent a modified version of the random-pattern task. Rather than acquiring recordings from a stationary site, here we surveyed the entire length of the STN by systematically moving the electrodes between task trials in small, discrete steps using automatic, computer-driven microdrive control (see Materials and methods, High-density STN survey).

SVR models for tremor and slowness were then calculated by incorporating recording data across all sites/trials within a single trajectory. Although decoding performance of models derived from multi-site data exhibited a trend of lower performance than models trained on single-site recordings (tremor: $r^2$=0.232±0.200 v. 0.073±0.052; slowness: $r^2$=0.125±0.108 v. 0.061±0.079, effective motor control: $r^2$=0.140±0.104 v. 0.043±0.058), these differences were not significant ($n$=203 stationary MER sites, $n$=17 moving MER trajectories, moving v. stationary data, LMM $\beta$=−0.075 to −0.075, $Z$=−0.945 to −1.291, $p$=0.197–0.344). Despite less data at each recording site, whole-STN models demonstrated above-chance decoding performance for all three metrics (tremor: 8/17 trajectories, slowness: 6/17, effective motor control: 6/17).

Recording sites along each trajectory were reconstructed using imaging (*Figure 4E*), and site-specific metric decoding $r^2$-values were calculated by applying the whole-STN SVR model to individual site recordings (*Figure 4F and G*) (see Materials and methods). Decoding performance was then compared across patients (*Figure 4H*). Across these recordings, tremor was optimally decoded at ($x = -13.0, y = -13.0, z = -5.0$; $Z = 1.911, p = 0.028$), while slowness was optimally decoded at ($x = -12.0, y = -15.0, z = -8.0$; $Z = 1.937, p = 0.026$). Within individual subjects, tremor was again found to be decoded dorsolaterally to slowness.

## Cortical recordings also revealed distinct representations of tremor, slowness, and effective motor control

Ten subjects additionally had ECoG recordings from sensorimotor cortex (motor cortex: $n = 16$ contacts, somatosensory cortex: $n = 15$, see Materials and methods). SVR models for metric decoding were similarly trained on ECoG signals. ECoG decoding performance did not differ between tremor or slowness ($n$=85 ECoG recordings across 27 sessions and 10 subjects, tremor: $r^2 = 0.323 \pm 0.153$, slowness: $r^2 = 0.314 \pm 0.143$, tremor v. slowness, LMM $\beta$=0.010, $Z$=0.578, $p = 0.563$).

To understand which spectral features contributed to cortical motor metric decoding, SVR model weights were aggregated across all patients and recordings and compared to metric-shuffled models. When compared directly, cortical tremor and slowness models had opposing relationships in $\alpha/\beta$ (8–40 Hz, $p$<0.027, permutation test), $\gamma_{mid}$ (45–125 Hz, $p$<0.023, permutation test), and $\gamma_{high}$ (150–225 Hz, $p$<0.015, permutation test) frequency bands (*Figure 5A*). Tremor models additionally had positive weights associated with $\theta$ frequencies (5–7 Hz, $p$=0.003, permutation test). Altogether, although cortical signals supported equivalent decoding performance for tremor or slowness, decoding features were nonetheless distinct. On the other hand, effective motor control decoding performance ($r^2 = 0.469 \pm 0.112$) was lower than tremor (effective motor control v. tremor, LMM $\beta$=−0.067, $Z$=−3.975, $p$=7.04*10^-5) and slowness (effective motor control v. slowness, LMM $\beta$=−0.057, $Z$=−3.397, $p$=0.001). Nevertheless, effective motor control was represented in cortical decoding models by $\gamma_{high}$ frequencies. These $\gamma_{high}$ features additionally appeared to differentiate effective motor control models from both tremor and slowness models (125–175 Hz, $p$<0.026, permutation test) (*Figure 5B–C*). In addition, $\alpha/\beta$

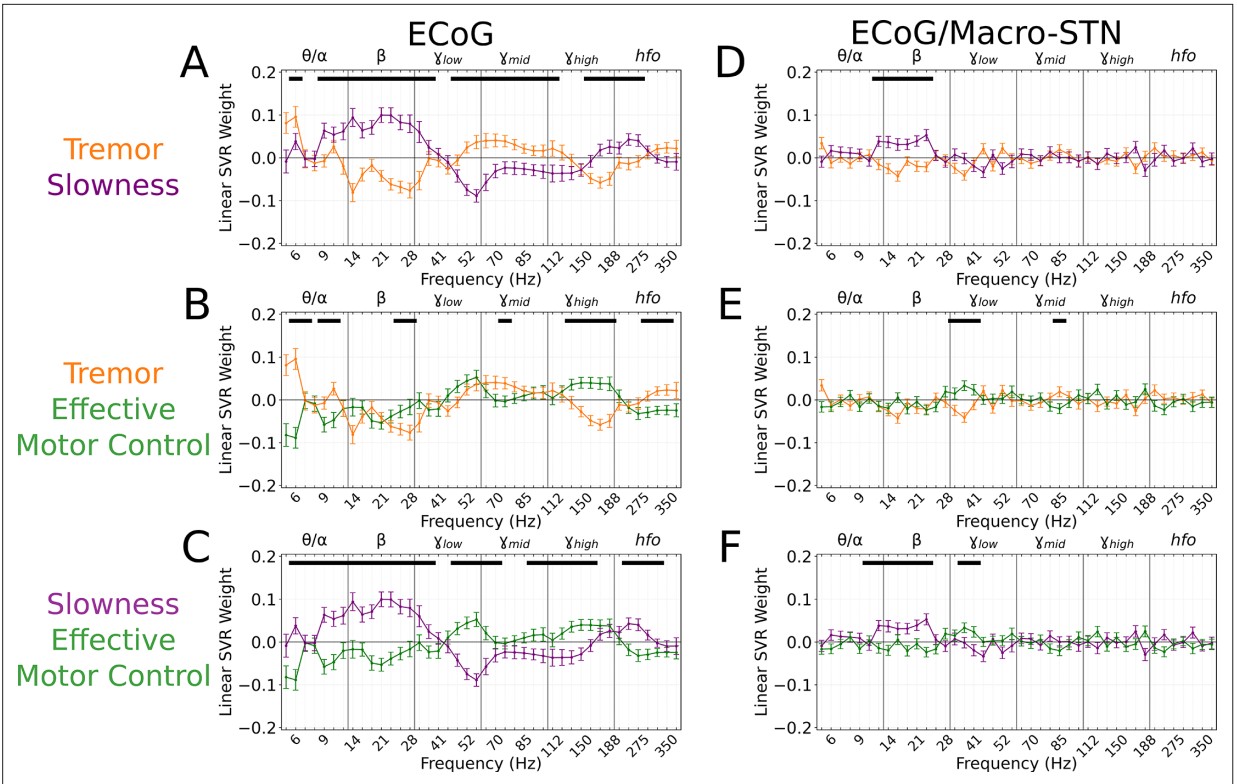

**Figure 5.** Cortical tremor and slowness decoding models exhibited opposing weights for multiple frequency bands, and co-expressed specific features with subthalamic recordings. (**A**) Average cortical tremor and slowness decoding model coefficients for every recording along sensorimotor cortex (*n*=85 electrocorticography [ECoG] recordings, 10 subjects). Colored lines indicate average weights, with positive/negative values reflecting a positive or negative relationship with the metric. Error bars indicate s.e.m. across subjects. Black lines (top) represented contiguous spectral features that significantly differed between tremor and slowness decoding models. (**B**) Average model coefficients for effective motor control and tremor. (**C**) Average model coefficients for effective motor control and slowness. (**D**) Average subthalamic nucleus [STN]-cortical coherence tremor and slowness decoding model coefficients for every pairwise recording along sensorimotor cortex and macro contacts within the STN (*n*=85 ECoG recordings, 10 subjects). (**E**) Average coherence model for effective motor control and tremor. (**F**) Average model coefficients for effective motor control and slowness.

(8–30 Hz, *p*<0.001, permutation test) and $\gamma_{low}$ (45–75 Hz, *p*<0.010, permutation test) frequencies exhibited an opposing relationship between effective motor control and slowness, much like the interaction between tremor and slowness models (*Figure 5C*). Taken together, although at different $\gamma$ frequencies, both STN ($\gamma_{low}$) and sensorimotor cortex ($\gamma_{high}$) exhibited features specific to effective motor control. And similarly to STN recordings, ECoG full-spectrum decoding was superior for tremor and effective motor control (full vs. beta-only decoding, LMM $\beta$=0.017–0.024, Z=2.224–2.577, *p*<0.026), but equivalent for slowness (full vs. beta-only decoding, LMM $\beta$=−0.013, Z=−1.531, *p*=0.126).

Finally, we analyzed whether motor features were selectively represented in different regions of cortex. ECoG recording sites (and their associated metric decoding performance) were plotted along a standard cortical surface (*Figure 6A*). Comparing tremor and slowness decoding performance by cortical anatomy revealed that slowness decoding had peaks in medial motor (*n*=31 ECoG recording sites, $x = −37.5, y = −15.8, z = +70.2$; *T*=2.210, *p*=0.030) and somatosensory ($x = −34.4, y = −37.4, z = +71.0$; $T = 2.340, p = 0.022$) cortices. We also observed a trending tremor decoding peak in lateral somatosensory cortex ($x = −50.2, y = −31.8, z = +64.0$; $T = 1.700, p = 0.093$) (*Figure 6B*). Similar to the STN, effective motor control decoding performance (relative to tremor or slowness decoding) was found not to differ by cortical anatomy (*p*>0.05). In sum, cortical signals supporting optimal tremor cortical decoding were found relatively lateral to those supporting better slowness decoding.

## Cortical decoding outperformed STN decoding

To understand whether motor (dys-)function was better represented in cortical signals, decoding performance was compared between patients with ECoG and STN recordings (*n*=10 subjects, 85

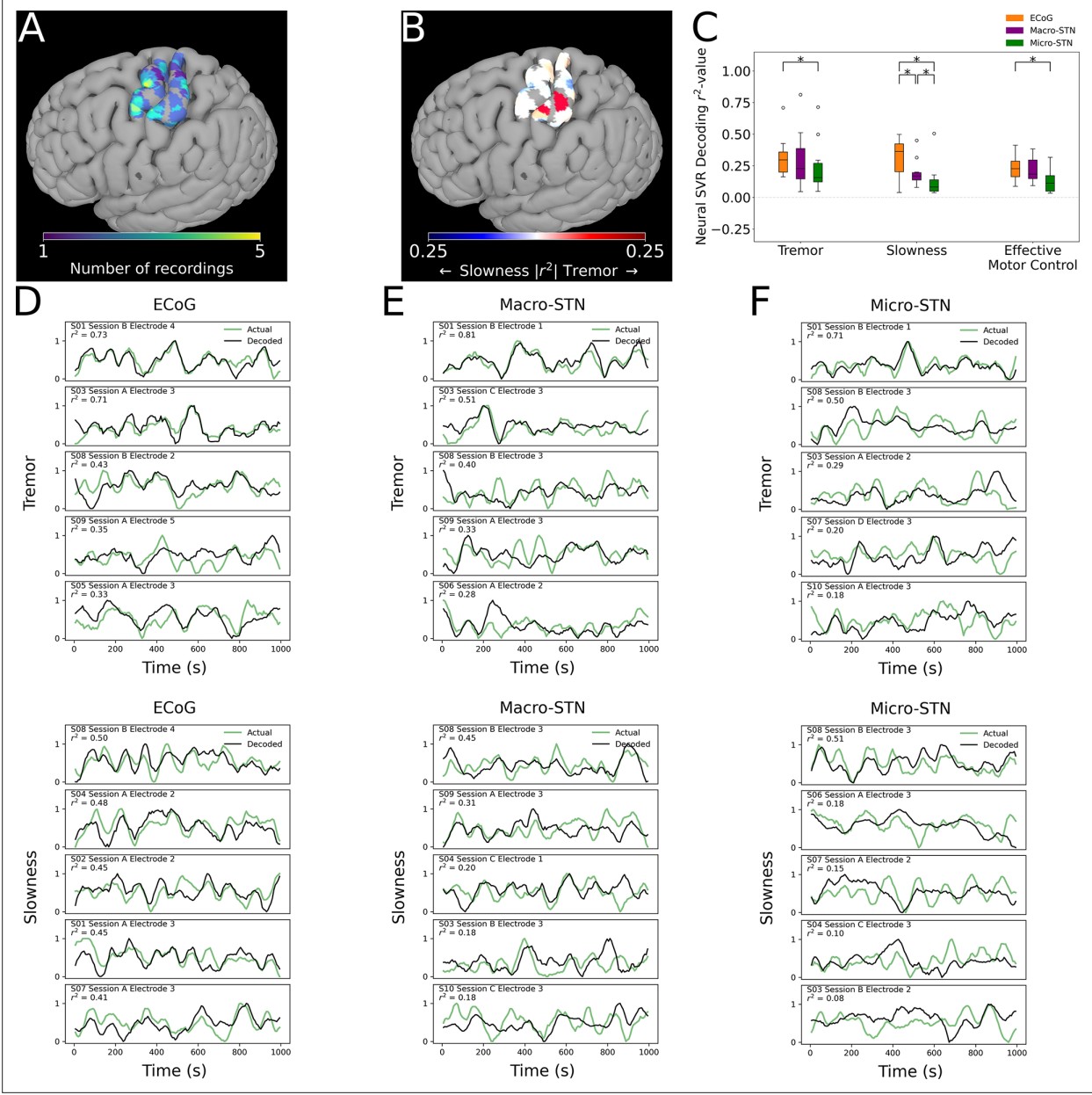

**Figure 6.** Cortical tremor and slowness decoding models were distributed throughout cortex, and generally were superior to subthalamic nucleus (STN) decoding models. (**A**) Recording density of electrocorticography (ECoG) contacts ($n$=31 ECoG sites, 10 subjects) on an MNI reference surface. (**B**) Difference in tremor vs. slowness decoding $r^2$-values for all cortical recordings. Warmer colors indicate surface vertices where tremor decoding was superior, whereas cooler colors indicate where slowness decoding was superior. (**C**) Decoding performance across metrics and recording types ($n$=85 ECoG recordings, 81 macroelectrode recordings, 81 microelectrode recordings, 10 subjects). Box represents interquartile range (25th–75th percentile), while whiskers indicate 5th–95th percentile of data range. Brackets indicate significant (*, $p$<0.05 in linear mixed model comparisons) differences in metric decoding $r^2$-values. (**D**) Examples of tremor (top) and slowness (bottom) decoding for top-performing ECoG contacts. Each row within the panel represents a different subject. (**E**) Examples of tremor (top) and slowness (bottom) decoding for top-performing macroelectrodes. (**F**) Examples of tremor (top) and slowness (bottom) decoding for top-performing microelectrodes.

ECoG recordings, 81 MER, 81 macroelectrode recordings) (*Figure 6C*). ECoG tremor decoding models exhibited higher performance than micro-STN recordings (ECoG v. micro-STN, $r^2$=0.323±0.153 v. 0.240±0.200, LMM $\beta$=0.047, $Z$=2.504, $p$=0.012), but only trended higher relative to macro-STN recordings (ECoG v. macro-STN, $r^2$=0.323±0.153 v. 0.294±0.222, LMM $\beta$=0.019, $Z$=1.001, $p$=0.317). Slowness decoding performance on the other hand was higher in ECoG models relative to both macro-STN (ECoG v. macro-STN, $r^2$=0.314±0.143 v. 0.195±0.105, LMM $\beta$=0.061, $Z$=4.445, $p$=8.78*$10^{-6}$) and

micro-STN (ECoG v. micro-STN, $r^2$=0.314±0.143 v. 0.128±0.134, LMM $\beta$=0.096, Z=7.025, p=2.14*10$^{-12}$) recordings. Like tremor, effective motor control ECoG decoding models exhibited superior decoding performance to micro-STN (ECoG v. micro-STN, $r^2$=0.233±0.105 v. 0.133±0.096, LMM $\beta$=−0.038, Z=−3.140, p=0.002), but not macro-STN (ECoG v. macro-STN, $r^2$=0.233±0.105 v. 0.219±0.098, LMM $\beta$=0.004, Z=0.312, p=0.755) recordings. In summary, recordings from sensorimotor cortex were superior to micro- and macro-STN recordings for decoding slowness, while cortical recordings were also superior to micro-STN recordings for decoding tremor and effective motor control. However, both cortical and deep recordings were able to meaningfully track patient symptoms (*Figure 6D–G*).

Because cortical and subthalamic decoding models used similar features for metric decoding, we investigated whether combining cortical-subthalamic recordings would be superior to using local signals. Pairwise coupling between cortical and macro-STN contacts was calculated via coherence within the same frequency range and sub-band features for SVR decoding. Across all metrics, local decoding models outperformed pairwise decoding models (tremor: ECoG v. EcoG/macro-STN, $r^2$=0.323±0.153 v. 0.075±0.029, LMM $\beta$=0.133, Z=15.073, p=2.43*10$^{-51}$; slowness: ECoG v. EcoG/macro-STN, $r^2$=0.314±0.143 v. 0.098±0.057, LMM $\beta$=0.123, Z=14.420, p=3.89*10$^{-47}$; effective motor control: ECoG v. EcoG/macro-STN, $r^2$=0.233±0.105 v. 0.096±0.050, LMM $\beta$=0.065, Z=10.213, p=1.73*10$^{-24}$). When examining decoding coherence features, slowness models positively weighed frequencies within the $\beta$ range (12–26 Hz, p<0.0005, permutation test) when compared to tremor and effective motor control decoding models. Effective motor control on the other hand positively weighed frequencies in the $\gamma_{low}$ range (34–45 Hz, p<0.010, permutation test) when compared to tremor and slowness decoding models. While tremor decoding models trended toward positive weights for $\theta$ (5–7 Hz, p<0.235, permutation test), these weights did not survive multiple comparisons. Overall, while cortical-subthalamic pairwise models relied on co-occurring physiological motifs, local neuronal information appeared more relevant to short-timescale decoding of motor dysfunction.

## Discussion

In this study we quantified PD tremor and movement speed in a structured motor task as surrogates for coarser clinical measurements of tremor and bradykinesia. We decoded these metrics using linear models and mapped these results to basal ganglia and cortical anatomy. We demonstrated that tremor and bradykinesia were represented by different functional motifs with distinct localization in the STN and sensorimotor cortex. We also contrasted these pathological states with periods of effective motor control, revealing unique markers of relatively symptom-free states. To our knowledge this is the first study to not only characterize the behavioral interaction between tremor and slowness within a single behavioral context but also to compare directly each motor sign's corresponding expression in neural activity, and further to compare this with relatively asymptomatic states. These results provide a holistic description of dynamic, spontaneous alternations in PD symptoms which reveal specific neurophysiological biomarkers of non-pathological and distinct pathological states.

We focused our neural decoding approach on two cardinal motor features of PD to isolate spectral features that reflected the expression of each. In the STN, tremor was characterized by lower-frequency ($\theta, \alpha$) oscillations in microelectrodes, whereas slowness was characterized by the presence of $\beta$ oscillations and the absence of $\gamma$ oscillations in macroelectrodes. Because $\gamma$ frequency oscillations are commonly associated with hyperkinetic states (*Lofredi et al., 2018*; *Swann et al., 2016*), our slowness decoding results may be understood in part as an 'anti-speed' neural model. Indeed, effective motor control was distinguished by $\gamma_{low}$ frequency activity, highlighting the importance of $\gamma$ frequency oscillations in effective movements. Some of these frequency bands in isolation ($\theta, \beta$) have been found to correlate with clinical measures of tremor and bradykinesia (*Asch et al., 2020*; *Neumann et al., 2016*; *Nie et al., 2021*). However, here we show directly the contrasting nature of distinct PD motor states both behaviorally and neurophysiologically, complement an evolving literature of dynamic STN states in PD (*Khawaldeh et al., 2022*), and highlight the dependence of these neurophysiological 'fingerprints' on the particular neural recording technique.

We also identified where tremor and slowness were optimally decoded (i.e., where metric-specific spectral information was greatest). Within our STN MER, optimal tremor decoding sites were found to be located within dorsolateral STN whereas optimal slowness decoding sites were more centrally located within the STN. Optimal tremor decoding may have included activity from zona incerta, a stimulation site commonly thought to be critical for alleviating tremor (*Plaha et al., 2008*; *Reck*

*et al., 2009*). Indeed, optimal stimulation sites to alleviate tremor and bradykinesia correspond to our dorsolateral-tremor/ventromedial-slowness topography (*Akram et al., 2017*). While several groups have localized $\beta$ frequency activity to dorsolateral STN, this has been observed to be located inferiorly to tremor-related higher frequency oscillations (*Tamir et al., 2020*; *Telkes et al., 2018*; *van Wijk et al., 2017*). Our tremor/slowness-dorsal/ventral STN results correspond to prior work suggesting subdomains within the STN. In macaques, motor cortex projects to the dorsal portion of the STN, and the ventral STN receives projections from prefrontal cortex (*Haynes and Haber, 2013*). In both macaques and humans, the ventral STN has been associated with stopping movement, while the dorsal STN is more associated with motor initiation and selection (*Chen et al., 2020*; *Mosher et al., 2021*; *Pasquereau and Turner, 2017*). In addition, the co-occurrence of tremor-frequency (4–8 Hz; $\theta$) activity in both STN and cortical recordings is consistent with previous work finding tremor-frequency oscillations ($\theta$) originating in the STN and propagating/synchronizing to motor cortex during tremor (*Hirschmann et al., 2013*; *Lauro et al., 2021*). Our anatomical results suggest that this propagation may be specific to dorsal STN. Slowness decoding models alternatively relied upon $\beta$ activity in macroelectrode recordings, perhaps reflecting anti-kinetic $\beta$ bursts relayed to ventral STN from inferior frontal or supplementary motor cortex (*Hannah et al., 2020*; *Oswal et al., 2021*). But while prior work did not directly compare the neuroanatomical substrate of distinct PD features within or across subjects, this work demonstrates how alternating motor features of PD may manifest along these anatomical subdivisions.

In general, cortical recordings were equally capable of decoding tremor or slowness and were generally superior to STN-based decoding as previously reported (*Merk et al., 2022b*). When comparing the feature weights of these decoding models, we observed opposing relationships in both $\beta$ and $\gamma$ bands. As previous studies have shown that tremor decreases $\beta$ oscillations across cortex (*Qasim et al., 2016*), and others have shown increased narrowband $\gamma$ activity during hyperkinetic/dyskinetic states (*Swann et al., 2016*), here we demonstrate a 'push-pull' relationship between these frequency bands in the alternating expression of tremor and slowness, and when comparing slowness and effective motor control models. While cortical $\beta$ oscillations (and their desynchronization with movement) are well characterized in PD (*Rowland et al., 2015*), the functional role of broadband $\gamma_{high}$/hfo activity is less clear. Although higher-frequency activity overlaps with phase-amplitude coupling peaks observed in cortex in unmedicated patients with PD (*de Hemptinne et al., 2013*), our models for effective motor control states suggested that $\gamma_{high}$ is specifically associated with more normal movement in line with previous research on human cortical sensorimotor mapping (*Crone et al., 1998*; *Fischer et al., 2017*).

With advances in technology, DBS aspires toward incorporating chronic neurophysiological recordings to help guide therapeutic stimulation (*Gilron et al., 2021*). While current closed-loop DBS paradigms trigger stimulation based on one or two frequency bands representing PD symptoms (*Kehnemouyi et al., 2021*), our results argue for the potential utility of a more targeted neurophysiological approach to PD state identification: more dorsal STN contacts may better sense signals reflecting tremor, while more ventral STN contacts may better identify signals corresponding to bradykinesia. Precise patient- and symptom-specific models could not only inform where to stimulate, but also when and *how* to stimulate (i.e., identifying stimulation settings to best treat tremor vs. bradykinesia). Most importantly, future neuromodulation paradigms could be derived not simply to disrupt pathological activity but actually to sustain the neurophysiological $\gamma$ frequency 'targets' of effective motor control. Looking ahead, chronic cortical recordings could work in concert with STN recordings to help identify precise motor states associated with specific aspects of disease expression much like prior work with essential tremor (*He et al., 2021*; *Opri et al., 2020*).

These results naturally suggest future prospective intra- or extra-operative studies applying these decoding models toward DBS control policies. While other studies decoding PD motor dysfunction from STN recordings have investigated the use of several decoding techniques (hidden Markov models, logistic regression, Kalman filters), their complexity/computational requirements will require balancing potential therapeutic benefits with the computational/power limitations of implanted pulse generators (*Hirschmann et al., 2017*; *Merk et al., 2022a*; *Shah et al., 2018*; *Yao et al., 2020*). Although relatively simple, our linear SVR approach was robustly able to fit patient- and symptom-specific decoding models in an interpretable and iterable fashion. Near-future studies using implanted hardware will require adaptation to device constraints, such as reducing the neural feature spectra

based on potentially reduced sampling rates (e.g., 3–100 Hz based on a device-based 250 Hz sampling rate), and to conform to potentially fewer available frequency band estimates, though future devices are expected to allow for greater flexibility in signal processing and algorithmic complexity.

While our motor metrics correlated with UPDRS subscores, we recognize that our single intraoperative behavioral task does not capture all aspects of PD motor dysfunction, and is not as naturalistic as other studies using chronic recordings (*Hirschmann et al., 2017*); nevertheless, our approach had the advantages of being objective, quantitative, and consistent, while providing the unique opportunity to compare different modalities of recordings (e.g., micro- and macroelectrodes). Further, because patients were withdrawn from dopaminergic medication for at least 12 hr prior to surgery, exacerbated patient symptoms in the operating room may produce improved model decoding performance relative to models trained on patients taking dopaminergic medication in the natural setting. Future studies implementing these models in the extra-operative and clinical setting may help bridge these gaps.

While we focused on spectral power measurements in each structure and coherence across structures, future work may examine whether instantaneous phase-based measures of synchrony across structures may potentially better decode motor states. Given the nonuniform spatial sampling of imaging-based reconstructions, our imaging-based analyses may have lacked sufficient power to reveal smaller-scale or additional neurophysiological-anatomical relationships. Because of our method of identifying intraoperative recording sites suitable for the task (identifying motor-responsive single units through clinical somatotopic testing), our recording site distribution and subsequent decoding sub-regions may have been biased toward dorsolateral recordings. As tremor decoding exhibited higher decoding performance than slowness, our observed slowness peak in ventral STN may reflect overall decreased decoding performance. In addition, as macro-STN recordings were necessarily 3 mm above MER, some recordings may have been collected outside of the STN. High-density STN MER may have additionally been impacted by larger macro-STN contacts deforming tissue in the initial downward recording trajectory. Although our decoding performance may be improved by larger datasets and/or more advanced machine learning approaches, our linear approach robustly achieved patient-specific decoding while revealing metric-specific neurophysiological signatures. Nevertheless, with our parametric measurements of PD motor behavior in the intraoperative setting, we were able to delineate the contrasting, push-pull relationship between neural states underlying tremor, bradykinesia, and effective motor control in both the STN and sensorimotor cortex.

## Materials and methods
### Study and experimental design

All patients undergoing routine, awake placement of deep brain stimulating electrodes for intractable, idiopathic PD between June 2014 and December 2018 were invited to participate in this study. Patients were selected and offered DBS by a multi-disciplinary team based solely upon clinical criteria (*Akbar and Asaad, 2017*). In this report, 32 subjects (*n*=27 subjects in stationary-recording experiments, *n*=5 subjects in high-density recording experiments) (4F, 28M; aged 47.5–78.5 years) underwent STN DBS. Subjects were off all anti-Parkinsonian medications for at least 12 hr in advance of the surgical procedure (UPDRS Part III: 48.3 ± 12.6). Twelve subjects were considered tremor-dominant, and 13 subjects had average tremor UPDRS III scores >2 in their dominant hand (*Jankovic et al., 1990*). Seventeen age-matched controls (14F, 3M; often patients' partners; aged 48.5–79.2 years) also participated in this study (patient v. control subjects ages, 65.2±7.9 vs 63.2±9.3, Mann-Whitney U-test, $p = 0.450$). Controls were required simply to be free of any diagnosed or suspected movement disorder and to have no physical limitation preventing them from seeing the display or using the manipulandum. There was a strong male bias in the patient population (2F, 25M) and a female preponderance in the control population (14F, 3M), reflecting weaker overall biases in the prevalence of PD and the clinical utilization of DBS therapy (*Accolla et al., 2007*; *Hariz et al., 2011*; *Rumalla et al., 2018*). Both patients and control subjects were predominantly right-handed (patients: 30 right-handed, 2 left-handed; control subjects: 16 right-handed, 1 left-handed). Patients and control subjects agreeing to participate in this study signed informed consent, and experimental procedures were undertaken in accordance with an approved Rhode Island Hospital human research protocol (Lifespan IRB protocol #263157) and the Declaration of Helsinki.

## Surgical procedure

MER from the region of the STN of awake patients are routinely obtained in order to map the target area and guide DBS electrode implantation. Microdrives (Alpha Omega Inc, Nazareth, Israel) were attached to a patient-customized stereotactic platform (STarFix micro-targeting system, FHC Inc) and then loaded with three parallel microelectrodes (*Konrad et al., 2011*). For 10 patients, ECoG strips were placed posteriorly along sensorimotor cortices through the same burr hole used for MER insertion to conduct intraoperative cortical recordings. The STN was identified electrophysiologically as a hyperactive region typically first encountered about 3–6 mm above estimated target (*Gross et al., 2006*). When at least one electrode was judged to be within the STN, electrode movement was paused and recordings were obtained in conjunction with patient performance of the visual-motor task.

## High-density STN survey

In five subjects, once the bottom of the STN was identified using typical electrophysiological procedures, custom-built routines using an FDA-approved software development kit (Alpha Omega, Inc) were used to automatically raise electrodes by a pre-specified distance between trials to conduct a high-density STN survey. To start, clinical MER was conducted in typical fashion. Once the electrodes were judged to have exited the STN, the length of the STN recording span was calculated based on intraoperative neurophysiology. Based upon this length, the electrodes were automatically raised by the microdrives in pre-calculated steps in coordination with the visual-motor task, during the inter-trial intervals. During this task, a separate control computer was used to coordinate the behavioral task with robotic control of the Alpha Omega neurophysiology and microdrive systems. Specifically, the FDA-approved C++ Neuro Omega software development kit was compiled into a custom Python library that could communicate with the Neuro Omega systems with a ~2 ms round-trip latency. From there, task-specific Python code enabled coordination with the behavioral control system. To acquire MER that spanned the STN, the length of the STN was estimated based upon standard neurophysiological assessment, and this length was divided by the number of task trials (typically 36). As the task was performed, the start of each inter-trial interval was detected by the control computer, and every few trials (typically 3), a command was issued to raise the electrodes by the appropriate distance. The task re-commenced once drive movement was complete (typically ~10 s later). This process continued until the subject completed the task and the microelectrodes had reached the top of the STN.

## Behavioral task and metrics

We employed a visual-motor target tracking task to estimate motor dysfunction in a quantitative and continuous fashion using MonkeyLogic (*Asaad et al., 2013*; *Asaad and Eskandar, 2008a*; *Asaad and Eskandar, 2008b*; *Hwang et al., 2019*). As per standard surgical procedure, patients were positioned in a reclined 'lawn-chair' position, supine on the operating table to maintain comfort while allowing patients to engage with the task and clinical assessment. For the task, a boom-mounted display was adjusted to the patient's line of sight, and patients were asked to verbally confirm their ability to see on-screen task objects. Patients using a joystick had it placed in their lap, while patients using a tablet had it placed on a stand in their lap (angle adjusted to comfort) and a stylus placed in their dominant hand. Healthy control subjects performed the task by sitting in a chair at a table, with the manipulandum secured on the table. Similar to the patients, the adjustable arm-mounted task display was adjusted to patient's line of sight. Subjects were instructed to follow a green target circle that moved smoothly around the screen by manipulating the joystick or stylus with their dominant hand with the goal of keeping the white cursor within the circle (*Figure 1A*). All subjects were instructed to not rest their dominant hand on their lap/the table while performing the task. The target circle followed one of several possible paths (invisible to the subject), with each trial lasting 10–30 s. Each session consisted of up to 36 trials (~13 min of tracking data).

Tremor amplitude was calculated from 3 to 10 Hz bandpass filtered cursor traces, $\left(TM_x\left(t\right) = \left|\left|C_{ax}\right|\right|\right)$. Movement speed was calculated from cursor traces lowpass filtered at 3 Hz to remove the influence of tremor, $\left(Speed_x = \left|\left|\frac{\Delta_x}{\Delta_t}\right|\right|\right)$. Both metrics were averaged into 7 s non-overlapping epochs to maintain consistency with our previous decoding approach (*Ahn et al., 2020*). To standardize movement speed within subjects, movement speed epochs within a session were min-max normalized into a measure of 'slowness,' where 0=highest speed and 1=lowest speed.

Effective motor control was quantified as the absence of tremor and slowness measures, relative to the entire session. Each epoch's 'effective motor control' measure was then calculated as $\frac{(1-Tremor)+(1-Slowness)}{2}$, where values of 0 indicated symptomatic states (tremor, slowness) whereas values of 1 indicated optimal motor performance.

Tremor and slowness were compared across control and PD populations using the following LMM: $y_{metric} = X_{population}\beta + Zu + \epsilon$, where $y_{metric}$ represented each epoch's metric amplitude and $X_{population}$ represented categorical labels of populations. LMMs were used to calculate the correlation between tremor and slowness across the entirety of each subject's behavioral data using the following model: $y_{tremor} = X_{slowness}\beta + Zu + \epsilon$, where $y_{tremor}$ represented each epoch's tremor amplitude and $X_{slowness}$ represented the epoch's simultaneous measurement of slowness.

To determine the timescale of metric fluctuation, autocorrelograms were calculated across each PD subject's behavioral data using 100 ms epochs. The average FWHM of the autocorrelograms were considered the minimum time necessary to label motor metric data as a 'symptomatic' period. Tremor or slowness were considered 'symptomatic' if they exceeded the 95th percentile of aggregate control data, and sustained symptomatic periods were defined as those persisting beyond the population metric FWHM continuously. For effective motor control, epochs were labeled 'symptomatic' if they were above the median of the PD subject's session distribution.

## LMM design

Behavioral metrics (tremor and slowness) were compared across control and PD populations using an LMM to account for each subject's asymmetric contribution of epochs: $y_{metric} = X_{population}\beta + Zu + \epsilon$, where $y_{metric}$ represented each epoch's metric amplitude, $X_{population}$ represented categorical labels of populations and associated fixed-effect regression coefficients ($\beta$), $Z$ represented the subject-specific random intercepts and their associated random effect coefficients ($u$), and $\epsilon$ represented the residuals. To understand the interactions and optimal timescales of tremor and slowness, each metric was calculated within smaller, 100 ms epochs. LMMs were used to calculate the correlation between tremor and slowness across the entirety of each subject's behavioral data using the following model: $y_{tremor} = X_{slowness}\beta + Zu + \epsilon$, where $y_{tremor}$ represented each epoch's tremor amplitude, $X_{slowness}$ represented the epoch's simultaneous measurement of slowness and the fixed-effect regression coefficient ($\beta$), and $Zu$ represented the same subject-specific random intercept design as above.

When assessing whether one type of metric (e.g., tremor) was preferentially decoded within a single type of recording (e.g., microelectrodes), SVR $r^2$-values were compared using the following LMM: $y_{r-value} = X_{metric}\beta + Zu + \epsilon$, where $y_{r-value}$ represented SVR decoding $r^2$-values from a single recording and metric, $X_{metric}$ represented categorical labels of metrics and associated fixed-effect regression coefficients ($\beta$), $Z$ represented the subject-specific random intercepts and their associated random effect coefficients ($u$), and $\epsilon$ represented the residuals. When investigating whether one type of recording was superior at decoding a single metric, $r^2$-values were compared using the following LMM: $y_r = X_{recording}\beta + Zu + \epsilon$, where $y_r$ represented SVR decoding $r^2$-values from a single recording and metric, $X_{recording}$ represented categorical labels of recording types and associated fixed-effect regression coefficients ($\beta$), and $Zu$ represented the same subject-specific random intercepts model as above.

When comparing cross-metric model performance (i.e., determining the ability of a model trained on tremor to decode slowness), performance was assessed by linear regression between the model's predicted metric (tremor) distribution and the co-occurring alternate metric (slowness) distribution. To compare the relative performance of tremor-trained models on decoding slowness, $r^2$-values were compared within recording type using the following LMM: $y_r = X_{metric}\beta + Zu + \epsilon$, where $y_r$ represented the SVR decoding $r^2$-value, $X_{metric}$ represented the categorical labels of either the model's trained metric (tremor) or the alternate metric (slowness) and their associated fixed-effect regression coefficients ($\beta$), $Z$ represented the subject-specific random intercepts and their associated random effect coefficients ($u$), and $\epsilon$ represented the residuals.

For datasets collected using the within-subject, high-density STN survey, SVR models were trained using recordings throughout the STN. Specifically, recordings at each depth were split into 2:1 train:test sets and aggregated for whole-STN SVR model fitting. $r^2$-Values from SVR models trained on high-density microelectrode data were compared to $r^2$-values from stationary microelectrode data using the following LMM: $y_r = X_{experiment}\beta + Zu + \epsilon$, where $y_r$ represented the SVR decoding $r^2$-value,

$X_{experiment}$ represented the categorical labels of experiment type and their associated fixed-effect regression coefficients ($\beta$), $Z$ represented the subject-specific random intercepts and their associated random effect coefficients ($u$), and $\epsilon$ represented the residuals.

## Neurophysiological signals and analysis

Microelectrode signals were recorded using 'NeuroProbe' tungsten electrodes (Alpha Omega), and macroelectrode signals were recorded from circumferential contacts 3 mm above the microelectrode tips. ECoG signals were acquired using Ad-Tech 8-contact subdural strips with 10 mm contact-to-contact spacing (Ad-Tech Medical, Racine, WI, USA). All signals were acquired at 22–44 kHz and synchronized using Neuro Omega data acquisition systems (Alpha Omega). Patients performed up to four sessions of the task, with microelectrodes positioned at different depths for each session. As microelectrodes were not independently positionable, some signals may have necessarily been acquired outside of the STN. All recorded signals were nevertheless considered and analyzed.

Neural data from the hemisphere contralateral to the patient's dominant hand were analyzed using the 'numpy/scipy' Python 3 environment (*Harris et al., 2020*; *Virtanen et al., 2020*). Offline, ECoG signals were re-referenced to a common median reference within a strip (*Liu et al., 2015*). All resulting signals were bandpass filtered between 2 and 600 Hz, and notch filtered at 60 Hz and its harmonics. These resulting timeseries were then downsampled to 1 kHz. Timeseries were bandpass filtered using a Morlet wavelet convolution (wave number 7) at 1 Hz intervals, covering 3–400 Hz. The instantaneous power and phase at each frequency was then determined by the Hilbert transform. To analyze broad frequency bands, we grouped frequencies into canonical ranges: $\theta/\alpha$ : 3–12 Hz, $\beta_{low}$ : 12–20 Hz, $\beta_{high}$ : 20–30 Hz, $\gamma_{low}$ : 30–60 Hz, $\gamma_{mid}$ : 60–100 Hz, $\gamma_{high}$ : 100–200 Hz, and *hfo* (high-frequency oscillations): 200–400 Hz. Power within the *hfo* band was interpreted as multiunit spiking activity, rather than discrete oscillations/ripples.

For cortical-subthalamic pairwise decoding models, neural synchrony was quantified using Welch's magnitude-squared coherence between timeseries within 7 s epochs (Hann windows, 1024 samples per segment with 512 sample shifts). The resulting coherence spectra were averaged into the same frequency bands as above.

## Imaging-based reconstruction of recording sites

Preoperatively, magnetic resonance (MR) images were obtained that included T1- and T2-weighted sequences (T1: MPRAGE, T2: SPACE; Siemens Vario 3.0T scanner). Pre-, intra-, and postoperative (in some cases) computed tomography (CT) scans were also acquired (Extra-Op CT: GE Lightspeed VCT Scanner; Intra-Op CT: Mobius Airo scanner). Postoperative T1-weighted MR images were typically obtained 1–2 days after the operation. To reconstruct recording locations, MR and CT images were co-registered using the FHC Waypoint Planner software and AFNI (*Cox, 1996*; *Li et al., 2016*). Microelectrode depths were calculated by combining intraoperative recording depth information with electrode reconstructions obtained from postoperative images using methods described previously (*Lauro et al., 2018*; *Lauro et al., 2016*). To determine the anatomical distribution of MER sites across patients, preoperative T1-weighted MR images were registered to a T1-weighted MNI reference volume (MNI152 T1 2009c) (*Fonov et al., 2009*). The resulting patient-specific transformation was then applied to recording site coordinates, which were then assessed for proximity to the STN as delineated on the MNI PD25 atlas (*Xiao et al., 2017*; *Xiao et al., 2015*; *Xiao et al., 2012*). ECoG contacts were segmented from intraoperative CT volumes, and were then projected onto individual cortical surface reconstructions generated from preoperative T1 volumes (*Dale et al., 1999*; *Fischl et al., 2002*; *Saad and Reynolds, 2012*; *Trotta et al., 2018*). Individual cortical surface reconstructions were co-registered to a standard Desikan-Destrieux surface parcellation (*Argall et al., 2006*; *Desikan et al., 2006*; *Destrieux et al., 2010*). Contacts within sensorimotor cortex (labeled as motor or somatosensory cortex by parcellation label – 'ctx_G_precentral' and 'ctx_G_postcentral', respectively) were considered for the present study.

## Neural decoding of behavioral metrics

To investigate whether STN or cortical activity could be used to estimate co-occuring behavioral metrics, SVR with a linear kernel using 'scikit-learn' was applied toward multi-spectral decoding of tremor or slowness (*Pedregosa et al., 2011*). Spectral power estimates for each canonical band ($\theta/\alpha$

, $\beta_{low}$, $\beta_{high}$, $\gamma_{low}$, $\gamma_{mid}$, $\gamma_{high}$, hfo) were further subdivided into 7 sub-bands for a total of 42 spectral features across 3–400 Hz (**Ahn et al., 2020**). SVR models trained on a single electrode's spectral features were fit using 100-fold Monte Carlo cross-validation with a 2:1 train/test split of temporal epochs within a task session. Model performance was assessed by linear regression (specifically, the squared Pearson $r$-value – $r^2$) between the observed and predicted metric distributions. To verify that these decoded results were not spurious, a separate set of SVR models were fit with a shuffled correspondence between behavioral metric data and neurophysiological signals in the training set.

When assessing whether one type of metric (e.g., tremor) was preferentially decoded within a single type of recording (e.g., microelectrodes), SVR $r^2$-values were compared using the following LMM: $y_{r-value} = X_{metric}\beta + Zu + \epsilon$, where $y_{r\text{-}value}$ represented SVR decoding $r^2$-values from a single recording and metric and $X_{metric}$ represented categorical labels of metrics. When investigating whether one type of recording was superior at decoding a single metric, $r^2$-values were compared using the following LMM: $y_r = X_{recording}\beta + Zu + \epsilon$, where $y_r$ represented SVR decoding $r^2$-values from a single recording and metric and $X_{recording}$ represented categorical labels of recording types.

When comparing the relative ability of a model trained on tremor to decode tremor or slowness, $r^2$-values were compared within recording type using the following LMM: $y_r = X_{metric}\beta + Zu + \epsilon$, where $y_r$ represented the SVR decoding $r^2$-value and $X_{metric}$ represented the categorical labels of either the model's trained metric (tremor) or the alternate metric (slowness).

Because these SVR models used a linear kernel, we extracted SVR model coefficients ('weights') to understand which spectral features were used to decode behavioral metrics. As linear SVR estimates of behavioral metrics ($Y_{behavior}$) are a combination of neural weights ($W_{neural}$) and power estimate ($X_{neural}$) inputs ($Y_{behavior} = W_{neural} \cdot X_{neural} + intercept$), positive weights described the association between the presence of a specific frequency band with higher metric output values. Conversely, negative weights described the absence of a neural feature when metric output values were high.

To test whether specific clusters of features ($\geq$3 contiguous spectral features) were consistently weighted across recordings, the distribution of each feature's SVR model weights (averaged over three adjacent features) across recordings were compared to the distribution of metric-shuffled SVR model weights using a contiguity-sensitive permutation test (**Ahn et al., 2020**). Over 10,000 iterations, each recording's SVR weight values were shuffled across the two models (empirical vs. shuffled), and the difference between individual feature distributions across electrodes was assessed using a paired $t$-test. The empirical $T$-statistic for each feature distribution (e.g., microelectrode $\beta_{1-3}$ empirical v. shuffled) was then compared to the null $T$-statistic distribution. The probability of the observed difference in $T$-statistics was calculated empirically from the null $T$-statistic distribution. This procedure was also used for understanding whether specific features were weighted differently for specific metric decoding models (e.g., microelectrode $\beta_{1-3}$ tremor v. slowness).

For datasets collected using the high-density STN survey, SVR models were trained using recordings throughout the STN. Specifically, data from each depth were split into 2:1 train:test sets and aggregated for whole-STN SVR model fitting. $r^2$-Values from SVR models trained on high-density microelectrode data were compared to $r^2$-values from stationary microelectrode data using the following LMM: $y_r = X_{experiment}\beta + Zu + \epsilon$, where $y_r$ represented the SVR decoding $r^2$-value and $X_{experiment}$ represented the categorical labels of experiment type.

To determine if whole-STN models could decode metrics above chance, $r^2$-value distributions from empiric and metric-shuffled decoding models were compared using the Wilcoxon test. To understand if specific recording sites contained information specific to individual metrics, metrics were estimated at each depth by applying the whole-STN SVR model weights to spectral features recorded at that depth. From there, $r^2$-values for each recording/depth were calculated between estimated and observed metrics.

## Anatomical analysis of metric decoding

To compare whether specific motor features were better decoded in different regions of the STN, tremor and slowness $r^2$-values were plotted in MNI coordinate space. All voxels and their associated $r^2$-values with MER recordings were then compared with a voxel-wise paired $t$-test with AFNI's '3dttest++' function. Each resulting voxel had an associated $Z$-statistic that was generated from 10,000 permutations of shuffling tremor and dataset $r^2$-values across voxels using AFNI's equitable thresholding and clustering (ETAC) algorithm (**Cox, 2019**). Briefly, ETAC was used to estimate

dataset-specific empirical statistical (e.g., *T*-statistic) and cluster-size (number of adjacent voxels) thresholds for significant results by running several (10,000) permutations of testing with shuffled data. However, due to the relatively low number of voxels per recording dataset (1 mm$^3$ per recording site compared to whole-brain coverage typically acquired with fMRI scans), no distinct clusters were isolated by the ETAC algorithm. For cortical surface-based comparisons between tremor and slowness $r^2$-values, '3dttest++' was also used, although without the ETAC algorithm as it is not currently implemented for surface-based datasets. Thus, instead of *Z*-scores computed by ETAC, we examined the *T*-statistic from '3dttest++' and its associated p-value. Colormaps for cortical figures were obtained from https://github.com/snastase/suma-colormaps ( *Nastase, 2018*).

## Statistical analysis

Data in text are represented as mean±standard deviation. All statistical tests, unless otherwise specified, were carried out in the 'scipy' environment. *p*-Values were adjusted for multiple comparisons wherever appropriate using the Benjamini-Hochberg procedure with $q = 0.05$ (*Benjamini and Hochberg, 1995*). Data points (epochs) were aggregated across trials within a single session. When comparing data aggregated across multiple subjects, LMMs were performed using the 'statsmodels' toolbox to disentangle the effect of interest (continuous or categorical) from the random effects/ unequal contributions of each subject's dataset (*Lindstrom and Bates, 1988*; *Seabold and Perktold, 2010*). In other words, individual epochs/trials/sessions/recordings were considered repeated measurements within individual subjects, and individual subject variability was considered as fixed factors during all statistical comparisons to avoid one subject unduly influencing results. All LMMs were random intercepts models, where each random intercept corresponded to a subject's dataset, and generally followed the formula of $y = X\beta + Zu + \epsilon$ , where $y$ represented the outcome variable, $X$ represented the continuous or categorical predictor variables, $\beta$ represented the fixed-effect regression coefficients, $Z$ represented the subject-specific random intercepts, $u$ represented the random-effects regression coefficients, and $\epsilon$ represented the residuals of the model fit. Once a model was fit, fixed-effect *p*-values were calculated from *Z*-scored parameter estimates (fixed-effect coefficients divided by their standard errors) against the normal distribution. When reporting LMM results, categorical comparisons were delineated as 'A v. B' whereas continuous regressions were noted as 'A × B.' When comparing $r^2$-value distributions with LMMs, descriptive statistics (mean±standard deviation) corresponded to distributions containing only each subject's highest-performing recording, while LMM statistics ($\beta$, $Z$, $p$) corresponded to comparisons of all recordings.

## Acknowledgements

We are grateful for the generous participation of our patients in this study. We thank Kelsea Laubenstein-Parker for technical assistance, Karina Bertsch for administrative support, and Ann Duggan-Winkle for clinical support. We also thank James Yu, Minkyu Ahn, David Segar, Tina Sankhla, and Daniel Shiebler for helping develop the motor task experiment. Finally, we thank Menem Andria and Alpha Omega for adding computer-controlled microdrive capabilities. National Institutes of Health Training Grant NINDS T32MH020068 (PML), Doris Duke Clinical Scientist Development Award #2014101 (WFA), National Institutes of Health COBRE Award: NIGMS P20 GM103645 (PI: Jerome Sanes) supporting WFA, Neurosurgery Research and Education Foundation grant (WFA), Lifespan Norman Prince Neurosciences Institute Brown University Robert J and Nancy D Carney Institute for Brain Science. Part of this research was conducted using computational resources and services at the Center for Computation and Visualization at Brown University, with funding provided by an NIH Office of the Director grant S10OD025181. WFA has received proprietary equipment and technical support for unrelated research through the Medtronic external research program.

## Additional information

### Competing interests

Peter M Lauro, Shane Lee, Daniel E Amaya, David D Liu, Umer Akbar: The authors have patents and patent applications (US patent #: 17312155, 17470710) broadly relevant to Parkinson's disease

(but not directly based upon this work). Wael F Asaad: The authors have patents and patent applications (US patent #: 17312155, 17470710) broadly relevant to Parkinson's disease (but not directly based upon this work). WFA has received proprietary equipment and technical support for unrelated research through the Medtronic external research program.

### Funding

| Funder | Grant reference number | Author |
| --- | --- | --- |
| National Institute of Neurological Disorders and Stroke | T32MH020068 | Peter M Lauro |
| Doris Duke Charitable Foundation | Clinical Scientist Development Award#2014101 | Wael F Asaad |
| National Institute of General Medical Sciences | P20 GM103645 | Wael F Asaad |
| Neurosurgery Research and Education Foundation | | Wael F Asaad |
| Lifespan Norman Prince Neurosciences Institute | | Shane Lee |
| Brown University Robert J. and Nancy D. Carney Institute for Brain Science | | Peter M Lauro |
| NIH Office of the Director | S10OD025181 | Wael F Asaad |
| Medtronic | | Wael F Asaad |

The funders had no role in study design, data collection and interpretation, or the decision to submit the work for publication.

### Author contributions

Peter M Lauro, Conceptualization, Formal analysis, Investigation, Methodology, Writing – original draft, Writing – review and editing; Shane Lee, Conceptualization, Supervision, Investigation, Methodology, Writing – review and editing; Daniel E Amaya, Investigation, Methodology, Writing – review and editing; David D Liu, Methodology, Writing – review and editing; Umer Akbar, Supervision, Investigation, Writing – review and editing; Wael F Asaad, Conceptualization, Supervision, Funding acquisition, Investigation, Writing – review and editing

### Author ORCIDs

Peter M Lauro ⓘ http://orcid.org/0000-0002-8569-6427
Wael F Asaad ⓘ http://orcid.org/0000-0003-4406-9096

### Ethics

Human subjects: Patients and control subjects agreeing to participate in this study signed informed consent, and experimental procedures were undertaken in accordance with an approved Rhode Island Hospital human research protocol (Lifespan IRB protocol #263157) and the Declaration of Helsinki.

### Decision letter and Author response

Decision letter https://doi.org/10.7554/eLife.84135.sa1
Author response https://doi.org/10.7554/eLife.84135.sa2

## Additional files

### Supplementary files
• MDAR checklist

### Data availability

The raw datasets supporting the current study contain patient information and are unique datasets under continued investigation for additional projects, including those of junior trainees. Deidentified

neural/behavioral estimates and related code to reproduce all analyses in the manuscript will be made available in a public repository (Dryad; https://doi.org/10.5061/dryad.h9w0vt4n4). To request raw datasets, please contact the corresponding authors (me@peterlauro.me, wael_asaad@brown.edu) with a project proposal. Based upon the granularity of the data requested and potential for patient information exposure, data sharing would be granted in consultation with the Lifespan IRB. There are no commercial restrictions for these data currently.

The following dataset was generated:

| Author(s) | Year | Dataset title | Dataset URL | Database and Identifier |
| --- | --- | --- | --- | --- |
| Lauro PM | 2023 | Data from: Concurrent decoding of distinct neurophysiological fingerprints of tremor and bradykinesia in Parkinson's disease | http://dx.doi.org/10.5061/dryad.h9w0vt4n4 | Dryad Digital Repository, 10.5061/dryad.h9w0vt4n4 |

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
