## [Editor Report]

This important study advances our understanding of Parkinson's by identifying micro and macro scale signatures linked to critical symptoms (e.g., tremor and slowness of movement), and effective motor control. The evidence supporting the conclusions is solid, and leverages a rich dataset obtained during naturalistic movement. The work will be of interest to neuroscientists, neurologists, and biomedical engineers.

---

## [Decision Letter]

**Decision letter after peer review:**

Thank you for submitting your article "Concurrent Decoding of Distinct Neurophysiological Fingerprints of Tremor and Bradykinesia in Parkinson's Disease" for consideration by *eLife*. Your article has been reviewed by 3 peer , one of whom is a member of our Board of Reviewing Editors, and the evaluation has been overseen by Tamar Makin as the Senior Editor. The following individual involved in the review of your submission has agreed to reveal their identity: Wolf-Julian Neumann (Reviewer #2).

Essential revisions:

Reviewers all agree that the work is interesting and beneficial, however, there are several aspects that the authors should address:

1) Clarifying how behavioural measures (tremor, slowness) were derived, and how these relate to clinical scores. The authors should also support their analysis choices such as normalisation of cursor speed and derivation of effective motor control;

2) Analysing sub-thalamic and cortical recordings together to explore connectivity and coupling measures;

3) Clarifying how multiple recordings from each subject have been dealt with for decoding and using R2 as a performance metric.

*Reviewer #1 (Recommendations for the authors):*

I enjoyed reading this manuscript and think it will be a valuable contribution to a number of research fields following a revision.

1) The manuscript would significantly improve if the authors provided additional information regarding their methods and reorganised their results.

A) How did the authors compute tremor amplitude probability densities in age-matched controls and patients who did not exhibit tremors? Panels 1B and C linked to tremor analysis are difficult to read – did most patients and controls have no tremor and therefore the density functions are decaying from 0 or is there a low amplitude peak?

B) What was the motivation for normalising cursor speed to its minimum and maximum and what are the implications of this normalisation when comparing speed within and across participants?

C) I am not entirely sure how useful age-matched control behaviours are to understanding the main results of the paper – the authors could consider removing these to streamline the Results section.

D) Could authors further clarify the analysis using FWHM to delineate periods of time where metrics were sustained above control levels?

E) the authors start referring to effective motor control in figure 2 but the description appears later in the paper; re-organising figures 2 and 3 would improve readability.

2) If ECOG and STN recordings were acquired simultaneously, how did signals in both structures co-vary? Would considering envelope – envelope, phase – phase, envelope – phase information improve decoding beyond what can be achieved from a single recording site?

3) In the discussion, the authors state that their model supports "tremor related oscillations originating in the STN and propagating to cortex" – what is the evidence for this in the manuscript?

4) Could the authors further discuss how their full spectrum decoder may be implemented in the future for DBS control taking into account device and real-time processing constraints?

5) Could the authors further discuss how tremor/slowness/effective movement decoding from micro-electrode recordings reflect overall activity levels of units (in particular those linked to higher frequencies (γ high and hfo)) (Figures 3 and 4)?

*Reviewer #2 (Recommendations for the authors):*

Thank you for inviting me to review this interesting study. This is an amazing paper, that I read with enthusiasm. I am not sure why the authors have chosen to neglect all brain signal decoding papers in the field of deep brain stimulation, perhaps they were afraid that this would diminish the novelty. I personally think that the paper and results are sufficiently novel and the paper would have further gained from a thorough discussion of the current field (for review from our group see Merk et al., Exp Neurol 2022; https://doi.org/10.1016/j.expneurol.2022.113993 ; I do not aim to get this paper cited, just want to provide some inspiration).

*Reviewer #3 (Recommendations for the authors):*

There are a few questions and suggestions that would strengthen the overall conclusions of the manuscript.

The approach they use to obtain high-density recordings of the STN involves first driving the microelectrode to the bottom of the STN, and then in an automated fashion and based on the length of the recording track, incrementally and automatically moving the microelectrode dorsally, interleaving experimental sessions at each increment, until the top of the STN is reached. This is a nice approach for mapping the entire STN. However, there are two questions. First, as far as I understand, the microelectrodes used will have a larger macro electrode contact 3mm dorsal to the microelectrode tip. This means that any tissue that lies 3mm to the most ventral aspect of the recording will be damaged by the larger macro contact. If the recording span of the STN is larger than 3mm, then some microelectrode recordings will be in this damaged region. How do the authors account for this, and would it make sense to discard data that is obtained >3mm dorsal to the dental STN border? Second, different recording tracks may have different lengths and spans. Which recording track is used for this process? As they note, not all the electrodes, therefore, recorded data from the STN, and so the question is whether it would make sense to discard these non-STN recordings.

Tremor amplitude is quantified as the magnitude of the 3-10 Hz filtered signal. How is this converted to a tremor score that is then used for effective motor control? In addition, effective motor control appears to be simply the average of the (effective) tremor and slowness scores. However, the simple average may be misleading as a very good score in one domain could potentially compensate for a poor score in the other. Had the authors instead considered using an effective motor control score that is the minimum of either the tremor or slowness effective scores (1 – their value)? On a related note, the authors convert cursor speed to slowness by normalizing within the session. But should they instead normalize within subjects?

Figure 2 could be presented more clearly. For example, there are no scale bars in 2A, and panels B, C, and D are all different sizes for example.

For figure 3, the authors use decoding models to compare the decoding of tremor to the decoding of slowness to determine which spectral features can distinguish between the two categories and between each category and effective motor control. This is a nice analysis. Interestingly, there is a difference in these features between micro and macro recordings, and the macro electrode features appear somewhat similar to the motor cortex ECoG recordings. Is there a reason this should differ from the micro features? More interestingly, is there a possible link between the macro STN recordings and the ECoG recordings? Have the authors investigated any measures of coupling or connectivity between the two regions?

There are multiple experiments performed at different depths throughout the STN in each subject. It is not clear to me, and I am sure the authors have addressed this point, but when constructing the LLMs, there is included a factor for the subject. However, are all experiments or trials recorded within the same subject considered as independent samples? Could one subject's data be driving these significant regression coefficients?

[Editors' note: further revisions were suggested prior to acceptance, as described below.]

Thank you for resubmitting your work entitled "Concurrent Decoding of Distinct Neurophysiological Fingerprints of Tremor and Bradykinesia in Parkinson's Disease" for further consideration by *eLife*. Your revised article has been evaluated by Tamar Makin (Senior Editor) and a Reviewing Editor.

The manuscript has been improved but there are some remaining issues that need to be addressed, as outlined below:

1) Could the authors please (a) clarify patients' and control subjects' limb position/posture during the behavioural task; (b) indicate that Louis et al. 2001 observed a relation between Parkinson's rest and action tremor when the UPDRS rest tremor sub-scores and the Washington Heights-Inwood Genetic Study of Essential Tremor Rating Scale were correlated; and (c) discuss why in this study there is a deviation between UPDRS rest tremor sub-score and tremor severity on task vs UPDRS kinetic tremor sub-score despite the previously reported relationship between the two (Louis et al. 2001).

---

## [Author Response]

Essential revisions:Reviewers all agree that the work is interesting and beneficial, however, there are several aspects that the authors should address:1) Clarifying how behavioural measures (tremor, slowness) were derived, and how these relate to clinical scores. The authors should also support their analysis choices such as normalisation of cursor speed and derivation of effective motor control;

We have sought to clarify the derivation of our motor metrics, and their relationship to UPDRS clinical assessments. In particular, we discussed the relationship of our tremor measurement with resting v. action/postural tremor – while our tremor measurement positively correlates with resting tremor, it trended towards positive correlation with action tremor as well. Because resting tremor is typically thought to be more specific to Parkinson’s disease (and our task uses continuous motion instead of static postures), we felt our task-based tremor measure was appropriate and relevant to PD.

Within-session speed normalization was used primarily to control for differing target speeds across different task trials and versions, while also accounting for variability in patient speed. To ensure this decision did not significantly alter our results, we now have confirmed that decoding slowness using within-subject or no speed normalization produced the same ultimate results (see details later in this response).

In addition, we have described the derivation of effective motor control earlier in the manuscript. When comparing our metric-average approach vs. the reviewer’s metric-minimum approach we saw equivalent decoding performance. We thus continue to report the same metric-averaged results as before.

2) Analysing sub-thalamic and cortical recordings together to explore connectivity and coupling measures;

We apologize for not investigating the (in retrospect) obvious question of combining subthalamic and cortical recordings for metric decoding. We quantified connectivity/coupling between electrodes using magnitude-squared coherence, and used the resulting spectra as neural features for decoding. Perhaps surprisingly, local decoding models out-performed paired-recording decoding models. However, when analyzing the relevant coherence decoding features, macro-STN and cortical coherence-based decoding relied on shared frequency bands (lines 402–418).

3) Clarifying how multiple recordings from each subject have been dealt with for decoding and using R2 as a performance metric.

We have clarified that for datasets compared across subjects using linear mixed models, individual epochs/trials/sessions/recordings are considered as repeated measurements for individual subjects, and individual subject variability was considered during all statistical comparisons to avoid one subject unduly influencing results. We describe our statistical approach further in the “Linear Mixed Model Design” sub-section of Materials and methods (including individual model/analysis design) in lines 646–690, and provide a brief overview of our approach in the “Statistical Analysis” sub-section of Materials and methods (lines 819–822).

All decoding-based results have been converted to r^2^, and the relevant statistics have been updated with our overall findings remaining intact.

Reviewer #1 (Recommendations for the authors):I enjoyed reading this manuscript and think it will be a valuable contribution to a number of research fields following a revision.1) The manuscript would significantly improve if the authors provided additional information regarding their methods and reorganised their results.A) How did the authors compute tremor amplitude probability densities in age-matched controls and patients who did not exhibit tremors? Panels 1B and C linked to tremor analysis are difficult to read – did most patients and controls have no tremor and therefore the density functions are decaying from 0 or is there a low amplitude peak?

Probability density plots were calculated in a similar fashion for all age-matched controls and patients regardless of the presence of tremor. Specifically, a normalized histogram of tremor amplitudes across all subjects and epochs was calculated. As patients with tremor were often not continuously exhibiting tremor throughout the task, the low-amplitude peaks for patient and control distributions likely represents a noise baseline within the 3–10 Hz band. The right-sided tail in the patient distribution, however, represents the wide range of tremor amplitude expressed by patients.

B) What was the motivation for normalising cursor speed to its minimum and maximum and what are the implications of this normalisation when comparing speed within and across participants?

Because the task’s moving target has a certain speed, this produces a sort of normalization in that the range of movement speed is at least somewhat related to that target speed in most instances. The practical effect of this is that normalization does not introduce a major change, and normalized speed produces the same ultimate results. When comparing speed across subjects and populations, speed was not normalized in order to understand if our task measurements of speed corresponded to bradykinesia in patients when compared to controls. Within individual subjects, speed was normalized in order to better identify when patients were slowing down relative to their own baseline.

Nevertheless we compared decoding r^2^ values across session-normalized and raw speed values, and found no difference in microelectrode recordings (session-normalized slowness vs. raw speed; r^2^ = 0.125 ± 0.108 v. 0.113 ± 0.094, LMM β = -0.004, Z = -0.741, p = 0.459), macroelectrode recordings (session-normalized slowness vs. raw speed; r^2^ = 0.198 ± 0.147 v. 0.181 ± 0.148, LMM β = 0.014, Z = 1.879, p = 0.060), and ECoG recordings (session-normalized slowness vs. raw speed; r^2^ = 0.314 ± 0.143 v. 0.317 ± 0.144, LMM β = -0.003, Z = -0.523, p = 0.601).

C) I am not entirely sure how useful age-matched control behaviours are to understanding the main results of the paper – the authors could consider removing these to streamline the Results section.

We included age-matched control subjects in the study to help validate our motor metrics as reflecting motor symptoms of PD (i.e. control behavior was used to establish cut-offs for pathologic behavior), across both fixed- and random-pattern versions of our task. In addition, the control data allowed us to characterize the distinct timescales of symptomatic tremor and slowness expression.

D) Could authors further clarify the analysis using FWHM to delineate periods of time where metrics were sustained above control levels?

We now realize this may have been unclear. As described in the Materials and methods section, to determine the timescale of metric fluctuation, autocorrelograms were calculated across each PD subject’s behavioral data using 100 ms epochs. The average full-width half-maximum (FWHM) of the autocorrelograms were considered the minimum time necessary to label motor metric data as a “symptomatic" period. Tremor or slowness were considered “symptomatic" if they exceeded the 95th percentile of aggregate control data, and sustained symptomatic periods were defined as those persisting beyond the population metric FWHM continuously. For effective motor control, epochs were labeled “symptomatic” if they were above the median of the PD subject’s session distribution.

E) the authors start referring to effective motor control in figure 2 but the description appears later in the paper; re-organising figures 2 and 3 would improve readability.

We apologize for the lack of clarity, in which effective motor control was introduced in Figure 2 before being introduced in the text. We have moved the text from the “Effective motor control had characteristic neural signatures” Results sub-section to the “Tremor and slowness were distinct and opposing symptomatic states” (lines 150–156).

2) If ECOG and STN recordings were acquired simultaneously, how did signals in both structures co-vary? Would considering envelope – envelope, phase – phase, envelope – phase information improve decoding beyond what can be achieved from a single recording site?

We thank the reviewer for suggesting this analysis. We analyzed synchrony between cortical and STN recordings by calculating magnitude-squared coherence between these timeseries within 7-second epochs. We then averaged the resulting coherence spectra into the same frequency bins as spectral power, and used these features for metric decoding. Perhaps surprisingly, local decoding models outperformed coherence decoding models for all metrics (lines 402–418). Nevertheless, we were able to observe shared neural features across ECoG and macro-STN recordings.

3) In the discussion, the authors state that their model supports "tremor related oscillations originating in the STN and propagating to cortex" – what is the evidence for this in the manuscript?

We interpreted the co-occurrence of tremor-frequency (4-8 Hz; θ) activity in both STN and cortical recordings as being consistent with tremor-frequency synchronization during tremor episodes, as described in work from our group (Lauro et al., 2021) and others (Hirschmann et al., 2013). We have since clarified the text in the discussion (lines 483–485).

4) Could the authors further discuss how their full spectrum decoder may be implemented in the future for DBS control taking into account device and real-time processing constraints?

Given the translational implications of our results, we appreciate the reviewer for the opportunity to discuss future considerations for implementing our full-spectrum decoder for future prospective experiments. We have added a paragraph to the discussion describing these considerations alongside other studies/techniques for decoding motor dysfunction in PD (lines 522–533).

Reviewer #2 (Recommendations for the authors):Thank you for inviting me to review this interesting study. This is an amazing paper, that I read with enthusiasm. I am not sure why the authors have chosen to neglect all brain signal decoding papers in the field of deep brain stimulation, perhaps they were afraid that this would diminish the novelty. I personally think that the paper and results are sufficiently novel and the paper would have further gained from a thorough discussion of the current field (for review from our group see Merk et al., Exp Neurol 2022; https://doi.org/10.1016/j.expneurol.2022.113993 ; I do not aim to get this paper cited, just want to provide some inspiration).Reviewer #3 (Recommendations for the authors):There are a few questions and suggestions that would strengthen the overall conclusions of the manuscript.The approach they use to obtain high-density recordings of the STN involves first driving the microelectrode to the bottom of the STN, and then in an automated fashion and based on the length of the recording track, incrementally and automatically moving the microelectrode dorsally, interleaving experimental sessions at each increment, until the top of the STN is reached. This is a nice approach for mapping the entire STN. However, there are two questions. First, as far as I understand, the microelectrodes used will have a larger macro electrode contact 3mm dorsal to the microelectrode tip. This means that any tissue that lies 3mm to the most ventral aspect of the recording will be damaged by the larger macro contact. If the recording span of the STN is larger than 3mm, then some microelectrode recordings will be in this damaged region. How do the authors account for this, and would it make sense to discard data that is obtained >3mm dorsal to the dental STN border? Second, different recording tracks may have different lengths and spans. Which recording track is used for this process? As they note, not all the electrodes, therefore, recorded data from the STN, and so the question is whether it would make sense to discard these non-STN recordings.

We agree that, in our experience, recordings “on the way up” where the microelectrode traverses a region already traversed by macroelectrodes, recording quality can be slightly diminished. However, it is still possible to observe single units which suggests that, to a useful extent, surrounding tissue may “collapse” around the smaller probe as it moves upward. Nonetheless, it is true that this tissue may be altered somewhat due to the earlier traversal, and so we now make mention of this as a potential limitation (lines 543–549).

Given the relative scarcity of data in the intraoperative setting, all recording tracks were considered. In the case of high-density STN survey data, we based drive movement on the “best” trajectory (i.e. longest span of representative STN neurophysiology and somatotopic responses). As you point out however, this means that some recordings may have been collected outside the STN. However, despite this, fixed-location recordings were able to decode well above chance, with macro-STN recordings (which would be more likely to be outside STN) demonstrating superior performance. Because of these shared limitations, we decided to include all high-density STN recordings.

Tremor amplitude is quantified as the magnitude of the 3-10 Hz filtered signal. How is this converted to a tremor score that is then used for effective motor control? In addition, effective motor control appears to be simply the average of the (effective) tremor and slowness scores. However, the simple average may be misleading as a very good score in one domain could potentially compensate for a poor score in the other. Had the authors instead considered using an effective motor control score that is the minimum of either the tremor or slowness effective scores (1 – their value)? On a related note, the authors convert cursor speed to slowness by normalizing within the session. But should they instead normalize within subjects?

Tremor amplitude was min-max normalized within each subjects from 0-1, which was subsequently used in the effective motor control equation.

We thank the reviewer for raising this important point of potentially obscuring motor dysfunction by the use of motor metric averaging. We re-calculated EMC using the method described by the reviewer, which was shown to have lower r^2^ values across micro-electrode recordings (average vs. minimum EMC; r^2^ = 0.140 ± 0.014 v. 0.092 ± 0.069, LMM β = -0.026, Z = -4.650, p = 3.32 * 10^-6^), but was equivocal for macro-electrode recordings (average vs. minimum EMC; r^2^ = 0.204 ± 0.097 v. 0.170 ± 0.116, LMM β = -0.014, Z = -1.837, p = 0.066) and ECoG recordings (average vs. minimum EMC; r^2^ = 0.233 ± 0.105 v. 0.245 ± 0.099, LMM β = 0.012, Z = 0.321, p = 0.748).

When comparing session- and subject-normalized slowness measurements, there was no difference in r^2^ values for micro-electrode recordings (session- v. subject-normalized slowness; r^2^ = 0.125 ± 0.108 v. 0.126 ± 0.105, LMM β = 0.001, Z = 0.195, p = 0.845), macro-electrode recordings (session- v. subject-normalized slowness; r^2^ = 0.198 ± 0.147 v. 0.197 ± 0.151, LMM β = 0.000, Z = 0.043, p = 0.966) and ECoG recordings (session- v. subject-normalized slowness; r^2^ = 0.139 ± 0.135 v. 0.140 ± 0.135, LMM β = -0.001, Z = -0.093, p = 0.926).

Because overall these metric modifications were mostly equivocal, we will continue reporting results with our original metrics.

Figure 2 could be presented more clearly. For example, there are no scale bars in 2A, and panels B, C, and D are all different sizes for example.

We have since updated all subpanels within Figure 2, and hope the results are more clear.

For figure 3, the authors use decoding models to compare the decoding of tremor to the decoding of slowness to determine which spectral features can distinguish between the two categories and between each category and effective motor control. This is a nice analysis. Interestingly, there is a difference in these features between micro and macro recordings, and the macro electrode features appear somewhat similar to the motor cortex ECoG recordings. Is there a reason this should differ from the micro features? More interestingly, is there a possible link between the macro STN recordings and the ECoG recordings? Have the authors investigated any measures of coupling or connectivity between the two regions?

We agree with the reviewer that micro-electrode and macro-electrode STN decoding models revealed intriguing differences in relevant neural features. We think the micro v. macro difference is due in part to the former recording type being less sensitive to local field potentials than the latter (Marmor et al., 2017). As for the similarities between macro-STN and ECoG recordings, this may be due to synchrony or volume conduction. As we re-referenced macro-STN and ECoG recordings in order to reduce volume conduction, shared neural features more likely represent functional coupling or shared representations of a motor state. It should be noted that the ranges of relevant frequencies in the β and γ bands differed between macro-STN and ECoG models, suggesting that there may be specific local features (particularly in sensorimotor cortex) in addition to shared motifs (Dubey and Ray, 2019).

However, to address whether these features represented metric-specific coupling between the STN and cortex, we calculated connectivity via magnitude squared coherence between these timeseries within 7-second epochs. We then averaged the resulting coherence spectra into the same frequency bins as spectral power, and used these features for metric decoding. Perhaps surprisingly, local decoding models outperformed coherence decoding models for all metrics (lines 402–418). Nevertheless, we were able to observe shared neural features across ECoG and macro-STN recordings.

There are multiple experiments performed at different depths throughout the STN in each subject. It is not clear to me, and I am sure the authors have addressed this point, but when constructing the LLMs, there is included a factor for the subject. However, are all experiments or trials recorded within the same subject considered as independent samples? Could one subject's data be driving these significant regression coefficients?

When comparing decoding accuracies across subjects with linear mixed models, individual decoding models from one session-recording (and their associated r or r^2^ values) are treated as one sample. As you mentioned however, subject is treated as a fixed factor to prevent one subject from skewing overall results. We describe our statistical approach further in the “Linear Mixed Model Design” sub-section of Materials and methods (including individual model/analysis design) in lines 646–690, and provide a brief overview of our approach in the “Statistical Analysis” sub-section of Materials and methods (lines 819–822).

[Editors' note: further revisions were suggested prior to acceptance, as described below.]The manuscript has been improved but there are some remaining issues that need to be addressed, as outlined below:1) Could the authors please (a) clarify patients' and control subjects' limb position/posture during the behavioural task; (b) indicate that Louis et al. 2001 observed a relation between Parkinson's rest and action tremor when the UPDRS rest tremor sub-scores and the Washington Heights-Inwood Genetic Study of Essential Tremor Rating Scale were correlated; and (c) discuss why in this study there is a deviation between UPDRS rest tremor sub-score and tremor severity on task vs UPDRS kinetic tremor sub-score despite the previously reported relationship between the two (Louis et al. 2001).

We apologize for not describing the subject/limb positioning previously, this had been eliminated to fit formatting requirements. We now describe these details in the “Behavioral Task and Metrics” sub-section of “Materials and methods” on lines 644–656:

“As per standard surgical procedure, patients were positioned in a reclined “lawn-chair” position, supine on the operating table to maintain comfort while allowing patients to engage with the task and clinical assessment. For the task, a boom-mounted display was adjusted to the patient’s line of sight, and patients were asked to verbally confirm their ability to see onscreen task objects. Patients using a joystick had it placed in their lap, while patients using a tablet had it placed on a stand in their lap (angle adjusted to comfort) and a stylus placed in their dominant hand. Healthy control subjects performed the task by sitting in a chair at a table, with the manipulandum secured on the table. Similar to the patients, the adjustable arm-mounted task display was adjusted to patient’s line of sight. Subjects were instructed to follow a green target circle that moved smoothly around the screen by manipulating the joystick or stylus with their dominant hand with the goal of keeping the white cursor within the circle (Figure 1A). All subjects were instructed to not rest their dominant hand on their lap/the table while performing the task.”

We now acknowledge the rating scales used to correlate resting and action tremor in the Louis et al. 2001 in text (lines 118–122):

“Although postural tremor can correlate with resting tremor when patients with PD are measured by the UPDRS for the former and the Washington Heights-Inwood Genetic Study of Essential Tremor (WHIGET) Rating Scale for the latter, resting tremor is thought to be more specific to the PD pathophysiology (Louis et al., 2001).”

Finally, there are several reasons to expect a complex relationship between UPDRS tremor scores (resting and kinetic) and performance on the target tracking task. First, we do observe (and now report) a relationship between resting and action/postural tremor UPDRS scores within our patient sample (Spearman ρ = 0.602, p = 0.002) in the text (lines 115–118), consistent with prior observations. Second, our task entails patient positioning that is different from that which is typically used to assess resting, postural, and action tremors, and is undertaken in the operating room environment under circumstances distinct from the standard in-clinic UPDRS evaluations. Third, there are likely aspects of the movements in our task that make it less directly comparable to resting, postural, or movement tremors at least as those are typically assessed. Importantly, we do observe a relationship between movement velocity and tremor on our task (Figure 2B, left), such that tremor is more strongly expressed at lower movement velocities, as one would expect for PD. Ultimately, the paramount consideration in our experiment was that neural activity should be directly compared against ground-truth behavioral metrics at identical, short timescales. One might argue that because our task did not fully recapitulate UPDRS findings in the clinic, the degree and type of tremor we observed during our experiment in the operating room may not fully reflect an individual’s level of motor dysfunction in a natural environment. Conversely, one might argue that because our task was naturalistic and required fine-motor goal-directed activity (in a way that is different from key aspects of the UPDRS exam), that we might have leveraged a behavior that is perhaps in some way more relevant to everyday function. In either case, it is highly unlikely our measure of tremor is irrelevant to the ground truth of an individual’s disease state. Most important, however these questions might be resolved, they do not diminish the value of an immediate, objective measure of tremor in order to understand its neurophysiological basis.